

# Photochemical aging of aviation emissions: transformation of chemical and physical properties of exhaust emissions from a laboratory-scale jet engine combustion chamber

Anni Hartikainen[1], Mika Ihalainen[1], Deeksha Shukla[2,3], Marius Rohkamp[4], Arya Mukherjee[1], Quanfu He[5], Sandra Piel[2,3], Aki Virkkula[6], Delun Li[6], Tuukka Kokkola[1], Seongho Jeong[3,4], Hanna Koponen[1], Uwe Etzien[2], Anusmita Das[3], Krista Luoma[6], Lukas Schwalb[3], Thomas Gröger[3,7], Alexandre Barth[8], Martin Sklorz[3], Thorsten Streibel[2], Hendryk Czech[2,3], Benedikt Gündling[4], Markus Kalberer[8], Bert Buchholz[2], Andreas Hupfer[4], Thomas Adam[4], Thorsten Hohaus[5], Johan Øvrevik[9,10], Ralf Zimmermann[2,3], 10 and Olli Sippula[1,11]

[1] Dept. of Environmental and Biological Sciences, University of Eastern Finland, FI-70211, Kuopio, Finland
[2] Analytical Chemistry, University of Rostock, D-18059 Rostock, Germany
[3] Helmholtz Zentrum München, D-85764 Neuherberg, Germany
[4] Dept. of Mechanical Engineering, University of the Bundeswehr Munich, D-85579 Neubiberg, Germany
[5] Institute of Climate and Energy Systems, ICE-3: Troposphere, Forschungszentrum Jülich, D-52425 Jülich, Germany
[6] Finnish Meteorological Institute, FI-00101 Helsinki, Finland
[7] Institute of Combustion Technology, German Aerospace Center (DLR), D-70569 Stuttgart, Germany
[8] Dept. of Environmental Sciences, University of Basel, 4056 Basel, Switzerland
[9] Norwegian Institute of Public Health, N-0213 Oslo, Norway
[10] Dept. of Biosciences, University of Oslo, N-0316 Oslo, Norway
[11] Dept. of Chemistry, University of Eastern Finland, FI-80101 Joensuu, Finland

*Correspondence to*: Anni Hartikainen (anni.hartikainen@uef.fi) and Olli Sippula (olli.sippula@uef.fi)

**Abstract.** Aviation is an important source of urban air pollution, but the impacts of photochemical processing on the exhaust 25 emissions remain insufficiently characterized. Here, the physical-chemical properties of fresh and photochemically aged emissions from a laboratory-scale jet engine burner operated with JP-8 kerosene were studied in detail with a range of online and offline methods. The fresh emissions contained high amounts of organic matter present predominantly in the gaseous phase. Photochemical aging in an oxidation flow reactor caused substantial formation of oxidized organic aerosol, increasing the particle mass by approximately 300 times. During aging, aromatic hydrocarbons and alkanes in the gas-phase decayed, 30 while gas-phase oxidation products, such as small carbonyls and oxygenated aromatics increased. The composition of organic matter became more complex by photochemical processing, with oxidation state increasingly growing throughout the addressed exposure range (equivalent to 0.2 to 7 days in atmosphere) with a $\Delta H{:}C/\Delta O{:}C$ slope of -0.54. Simultaneously, the near-UV wavelength absorption by the particles increased due to enhanced particulate mass. The imaginary refractory indices of organic particulate matter were 0.0071 and 0.00013 at the wavelength of 520 nm for the fresh and photochemically processed 35 particles, respectively, indicating secondary production of weakly absorbing brown carbon. The direct radiative forcing by the exhaust particles was estimated by a Mie-model, which revealed a prominent shift from warming to cooling climate effect



upon photochemical aging. The results highlight the importance in considering secondary aerosol formation when assessing the environmental impacts of aviation.

## 1 Introduction

Particulate emissions from aircrafts have been associated with deteriorating urban air quality near-source or downwind airports, contributing to adverse health effects and changes in direct and indirect climate forcing (Lee et al., 2023; Riley et al., 2021; Yim et al., 2015). There is a concern that the health impacts of aviation may be enhanced by the small size of the emitted particles, as they are principally ultrafine particles (UFP) with diameter below 100 nm, capable of depositing in the peripheral airways and translocate into circulation (Ohlwein et al., 2019; Schraufnagel, 2020). Based on ambient observations, particles

from aircraft exhausts can be distinguished from the background urban air specifically by their small size, and they often dominate the particle number concentration in near-airport areas (Hudda et al., 2016; Masiol et al., 2017; Stacey, 2019; Zhang et al., 2020).

In aircraft engines, particulate matter (PM) emissions are formed in the combustor mainly as non-volatile, carbonaceous mature soot and from condensation of organic vapors when the exhaust cools at the engine exit (Masiol &

Harrison, 2014; Vander Wal et al., 2014). The primary emission formation process is governed by engine conditions with typically high contribution of elemental carbon (EC) at higher loads, while at lower engine loads, including idling and taxiing on the airport, the particles are characterized by higher organic carbon (OC) content (Delhaye et al., 2017; Elser et al., 2019; Presto et al., 2011; Vander Wal et al., 2014). Further, fuel choice and engine design can have markedly high impacts on the emission contents (Durdina et al., 2021; Kelesidis et al., 2023; Kilic et al., 2017; Rohkamp et al., 2024; Schripp et al., 2022).

In addition to carbon dioxide ($CO_2$) and carbon monoxide (CO), the gaseous emissions from aircrafts include nitrogen oxides ($NO_x$; NO and $NO_2$), sulfur dioxide ($SO_2$), and a range of organic gaseous compounds. In the atmosphere, the emitted gases may react with atmospheric oxidants, which results in formation of lower-volatility products and consequently airborne secondary particles. Previous estimates of the secondary particle formation from aircraft engines are scarce but point towards significant secondary organic aerosol (SOA) formation. SOA mass may exceed the amounts of fresh particulate matter by one

or two orders of magnitude at low load operation (Kiliç et al., 2018; Miracolo et al., 2011, 2012). There is, however, a lack of detailed chemical-physical characterization of the produced secondary particles, which would be essential for assessing their environmental health and climate effects.

Aviation emissions influence radiative forcing in the atmosphere directly by absorbing or scattering sunlight, and indirectly, mainly by stratospheric contrail formation (Lee et al., 2021). Black carbon (BC) containing soot essentially

composed of graphite-like EC structures absorbs sunlight efficiently with little wavelength dependency in the visible and near-visible spectrum (Michelsen et al., 2020). In addition, organic aerosol particles can absorb light in the ultraviolet (UV) range and shorter visible wavelengths. This absorption by the organic 'brown carbon' (BrC) has typically high wavelength dependency and composition- and size-dependent refractive index (Andreae & Gelencsér, 2006; Saleh et al., 2018). Ultimately,



the direct radiative forcing caused by aviation particles is impacted by both the primary BC and the potentially BrC-containing

organic aerosol formed during the exhaust cooling or in the atmosphere. Contribution of SOA to the radiative forcing efficiency of aviation exhausts is, however, not included in current estimates. Especially the compositions of long-range transported, atmospherically aged exhaust emissions remain poorly described, even though they may substantially contribute to the radiative forcing in the atmosphere as well as to the health-related impacts of aerosols near airports.

        This work aimed to examine the impacts of photochemical aging on the chemical, physical, and optical properties of

small-scale jet engine exhaust emissions. The influence of photochemical processing on the properties of jet engine exhaust was gauged by simulating photochemical aging in an oxidation flow reactor (Photochemical Emission Aging flow Reactor (PEAR); Ihalainen et al., 2019). Online and offline analyses of gaseous and particulate species were combined to characterize the impacts of photochemical processing aviation emissions. Finally, the radiative forcing efficiencies of the fresh and aged exhaust particles were estimated by Mie-modelling of the radiative forcing based on the measured aerosol properties.

## 2 Methodology

### 2.1 Operation of the laboratory-scale jet engine combustion chamber

The experiments were performed with a combustor test rig consisting of an air mass flow control unit, air supply plenum, fuel control unit, combustion chamber, and a measuring area with installed pressure, temperature, and gas probes. An original combustion chamber taken a small jet engine with a net thrust of 180 N was used (Hupfer et al., 2012). The rig was operated

under ground-level atmospheric conditions. No lubrication oil was included in the system, so that the secondary aerosol formation from kerosene combustion could be studied separately.

        The fresh gaseous and particulate emissions at various settings were compared to the literature values available at the ICAO Aircraft Engine Emission Databank (International Civil Aviation Organization, 2022). A stable load of 7 % was selected as representative of an average Landing and Take-Off cycle (LTO) of a commercial aircraft, based on trial runs at different

air-fuel ratios (AFR) and flow speeds. The current aviation regulations provide emission limits for the total LTO cycle on average, further justifying the use of a single representative load for the assessment of atmospheric processing of the exhaust emissions. AFR during the stable operation of the combustor rig was $85 \pm 0.6$ (g s$^{-1}$ per g s$^{-1}$; arithmetic mean $\pm$ standard deviation of means of eight runs, four hours each). More information on the test rig operation conditions is available in Fig. S1.

The fuel in use was kerosene-based tactical air fuel, specified according to Jet A1 or JP-8. Fuel of same quality from two barrels was used during the campaign. The jet fuel was analyzed by two-dimensional gas chromatography (GC) coupled to high resolution time-of-flight mass spectrometry (see Supplementary section S1.1 for analysis details). Average composition of $C_{14}H_{26}$ was determined based on the high-resolution mass spectrometry results. The jet fuel was primarily (57 %) composed of aliphatic hydrocarbons, but low alkylated one- and two-ring aromatic structures were also detected (27 % benzenes, 11 %

naphthalenes, and 5 % biphenyles). Higher polyaromatic hydrocarbons (PAHs) were not detected. The lack of oxygenated





species as well as the large share of saturated hydrocarbons in the fuel is evident in the Van Krevelen plot of the elemental composition of mass fragments detected in the fuel mass spectra (Fig. 1a).

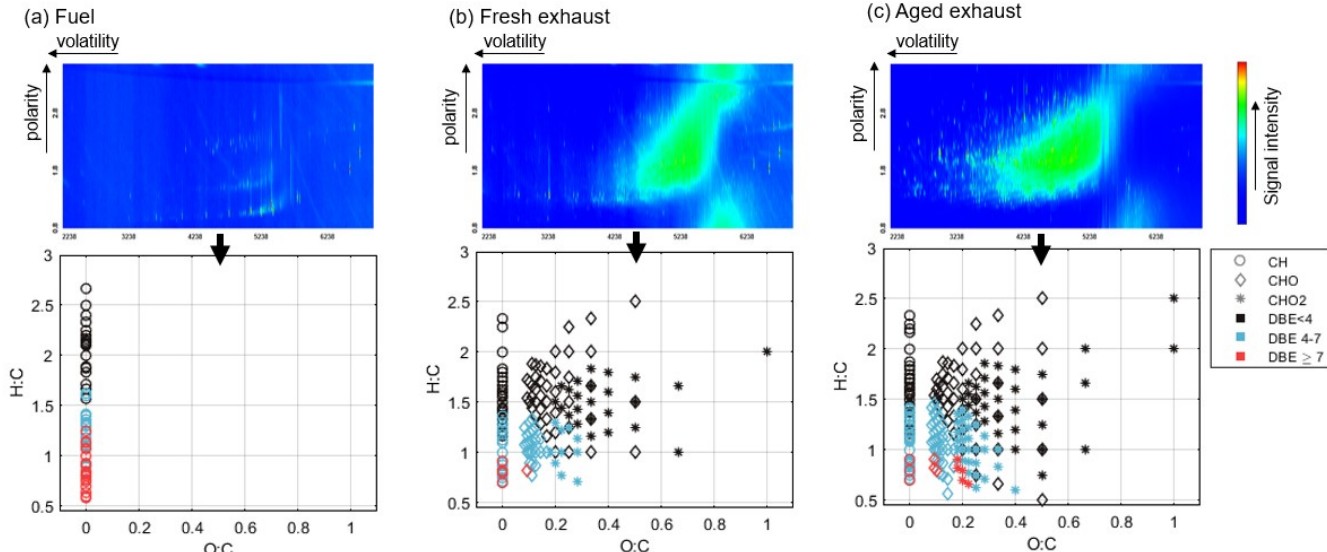

**Figure 1. GC×GC-MS chromatographs and respective Van Krevelen plots of the elemental ratios of mass fragments for the thermally accessible organic particulate matter in the fuel (a), fresh exhaust (b), and aged exhaust (c). Please note that GCxGC plots are for qualitative purposes only and that the intensity data may not correspond to the mass composition in the emission. DBE of 4 corresponds to core structures with one aromatic ring, whereas DBE ≥ 7 indicates two-ring aromatic structures.**

## 2.2 Sampling and dilution

The experimental setup is illustrated in Fig. 2. Exhaust from the jet engine burner was sampled to the aerosol instrumentation through a heated precyclone (400 °C) and a heated line (350 °C) and diluted by combination of a porous tube diluter (PTD) and ejector diluter (ED) (DAS, Venacontra, Finland) to minimize particle losses during sample cooling. The dilution was controlled by an automated dilution system, based on continuous measurements of $CO_2$ concentrations in the sample before and after the dilution and in the clean air used for dilution. Four hour runs (n = 3 for fresh, n = 4 for aged exhaust) were performed at dilution ratio (DR) of 50 ± 0.04. DR was selected with the aim of reaching suitable doses for concurrent cell exposure experiments, depicted in a separate study. The impact of different extents of photochemical processing were additionally assessed in experiments conducted at DR of 200. In the aging experiments, an additional dilution (DR of 10) was performed with an ED (Palas GmbH, Germany) prior to the online particle phase measurements due to the high PM concentrations in the photochemically aged exhaust.



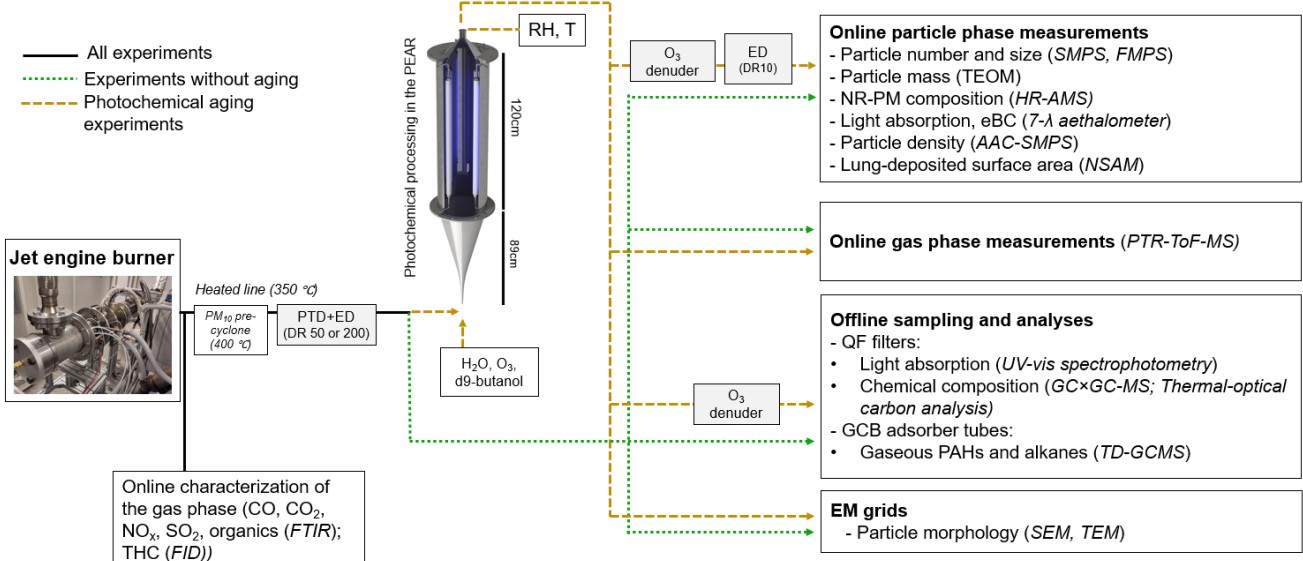

**Figure 2. Schematic of the experimental setup and characterized aerosol properties. The instrument or analysis method in use for assessment of each emission characteristic is shown in parentheses.**

Filter samples of the fresh and aged particles were collected on quartz fiber filters (QFF, 47 mm, Whatman QM-A, cytiva). QFFs were baked for 5 h at 550 °C prior to sampling to remove possible contaminants. Partisol Model 2300 Speciation Sampler (Rupprecht & Pataschnick Co., Inc., USA) with a sharp-cut $PM_{2.5}$ impactor was operated during the 4 h runs at either 16.7 or 8.35 l min$^{-1}$. Gas-phase organics of different volatilities were collected on adsorber tubes with three sublayers of GCB (Graphitized Carbon Black) sorbents (Table S1; Supplementary section 1.2). Adsorber tubes were conditioned under a protective nitrogen atmosphere at 350 °C for 1 h 30 min. QFFs were installed in front of the adsorber tubes to remove particle fractions. Gillian GilAir Plus sampling pumps (Gilian, USA) were used for sampling for 240 min at a flow rate of 0.5 l min$^{-1}$ accounting for a total volume of 120 l. The aged exhaust was passed through ozone denuders upstream to the particle/gas phase offline sampling. The denuders consisted of ceramic honeycombs which were impregnated with potassium nitrite to decompose ozone in the sample. Both particle and gas phase samples were stored in the freezer at -20 °C until analyses were performed.

## 2.3 Photochemical processing

Photochemical processing was performed using the high-volume Photochemical Emission Aging flow Reactor (PEAR; Ihalainen et al., 2019). The sample travel time from the test rig to the PEAR was estimated to be 3 s. The flow rate through the PEAR was 100 l min$^{-1}$, resulting in a mean residence time of 70 s for the exhaust sample in the reactor. Relative humidity after the PEAR was 47 % ± 1.6 % and temperature was 23 ± 0.9 °C. The PEAR contains four UV (254 nm) lamps, output of which can be controlled individually by 1 to 10 V analog signals. The extent of photochemical process was monitored using d9-butanol as an "hydroxyl radical (OH·) clock" (Barmet et al., 2012). OH· exposures are presented as atmospheric equivalent



days (eqv.d) at the ambient average OH· concentration of $1.5 \times 10^6$ molec. cm$^{-3}$. A range of photochemical processing (0.5 to 7 eqv.d; Table S2), was examined in the DR200 experiments by use of selected lamp settings. For the 4h experiments, d9-butanol was not input alongside the exhaust sample to avoid impacts with the concurrent cell exposures, not depicted here. Instead, the

exposure (approx. 2 eqv.d) was assessed in separate experiments in the same oxidation conditions.

External OH reactivity (Peng & Jimenez, 2020) of the sample was estimated to be 280 s$^{-1}$ or 70 s$^{-1}$ for the DR50 or DR200 experiments, respectively. The ratio of photolysis to OH exposure remained between 0.6 and $2 \times 10^6$ cm s$^{-1}$ (Table S2), indicating that in total, the flow reactor conditions were in the 'risky' regime as defined by Peng and Jimenez (2017). This means that major non-tropospheric photolysis was prevented by the careful conditioning of the sample and flow reactor

conditions, but the photolysis was slightly higher compared to tropospheric conditions, which may have influenced the formation of the oxidation products. The particle wall losses in the PEAR are minimized by the high volume-to-surface ratio, optimized laminar flow profile and by disregarding the flow stream closest to the walls from the successive sample (Ihalainen et al., 2019). The particulate condensation sink grew from approximately 7 s$^{-1}$ to over 400 s$^{-1}$ by the secondary aerosol formation, and simultaneously the lifetime of low-volatility organic compounds with respect to particulate condensation

decreased from 0.1 s to 0.002 s. Thus, any LVOCs can be expected to have condensed onto the particles within the PEAR residence time. See Supplementary section 2 for details of PEAR condition characterization.

**2.4 Gas phase analyses**

The offline analysis of alkanes and aromatic compounds in fresh and aged exhaust gases was performed on the GCMS-QP2010 Ultra (Shimadzu, Japan) equipped with the thermal desorber unit (TD-20, Shimadzu). An isotope-labelled standard mixture

was applied on the adsorber tubes prior to analysis and equivalent target compounds and similar target groups were quantified according to respective standard compounds as documented in Table S3. Details of the offline gas phase analyses are available in Supplementary section S1.2.

Concentrations of CO, CO$_2$, NO$_x$, and SO$_2$ in the raw exhaust gas were measured directly through a heated line by a Fourier-transform infrared gas analyzer (FTIR; DX4000, Gasmet, Germany). Additionally, the THC content in the undiluted

exhaust was assessed by a flame ionization detector (FID, Thermo-FID, SK-Elektronik, Germany) calibrated daily with a 30 ppm propane calibration gas. Volatile organic compounds (VOCs) in the diluted fresh or aged exhausts were measured online by a proton transfer reactor - time of flight - mass spectrometer (PTR-ToF-MS; PTR-TOF 8000, Ionicon, Austria) in a time-resolution of 10 s. Hydronium ions were used as the primary reagent, so that compounds with proton affinity higher than that of water (691 kJ mol$^{-1}$) were protonated and measured in high spectral resolution in ppb-level. In other words, gaseous aromatic

hydrocarbons, alkenes, oxidized VOCs, and larger ($>C_{10}$) alkanes were quantified by the PTR-ToF-MS, but for example small alkanes were not captured. PTR-ToF-MS was operated at the E/N-ratio of 137 Td. The PTR-ToF-MS peak table and reaction rates applied for the emission index determination are available in Table S4. The ions measured by the PTR-ToF-MS were classified based on the resolved molecular formulas.



## 2.5 Particle size distribution and density measurements

The particle size and number were measured by a scanning mobility particle sizer (SMPS; Classifier Model 3082, DMA Model 3081, TSI, USA) and a fast mobility particle sizer (FMPS; Model 3091, TSI, USA) in the respective size ranges of 7 – 305 nm and 6 – 560 nm and time resolutions of 3 min and 1 second. The total particle mass was directly measured with a tapered element oscillating microbalance (TEOM; Model 1400a, Thermo Fisher Scientific, USA). The TEOM inlet tube was heated to 50 °C leading to the possible evaporation of the semi-volatile fraction of the PM. Thus, the TEOM-based particle masses

($PM_{TEOM}$) should be taken as the lower limit of the total particle mass. The lung-deposited surface area (LDSA) of the particles was measured by a nanoparticle surface analyzer (NSAM; Model 3550, TSI, USA). The NSAM was set to quantify LDSA in the alveolar region of the human lung.

Particle density was measured by an aerodynamic aerosol classifier coupled with scanning mobility particle sizer (AAC-SMPS; Combustion Ltd., UK- Model 5420, Grimm Aerosol Technik, Germany). The AAC was set to target

aerodynamic size modes of 20, 40, and 80 nm (in fresh exhaust) or 20, 40, 80, and 160 nm (in photochemical experiments). The particle effective density ($\rho_{eff}$) at the selected sizes was estimated based on the subsequent mobility-sized based separation of the particles by an SMPS (Allen & Raabe, 1985; Zhai et al., 2017). The particle density in the photochemically processed aerosol was estimated also by comparing the particle mode aerodynamic diameter measured by the high-resolution aerosol mass spectrometer (HR-ToF-AMS, Aerodyne Research Inc., Billerica, MA) to the mobility mode diameter obtained by SMPS.

SMPS-based PM mass estimations ($PM_{SMPS}$) were determined from the SMPS size distributions by applying the resolved effective densities, namely, 1.35 g cm$^{-3}$ for both fresh and aged exhaust particles.

Particles were also sampled for electron microscopy on carbon holey grids (Agar Scientific, S147-400) using an aspiration sampler (Lyyränen et al., 2009). Sampling was done from the diluted sample at a flow rate of 0.3 l min$^{-1}$. The morphology of the particles was then examined by scanning electron microscopy (SEM, Sigma HD/VP; Carl Zeiss NT) to

visualize the morphology of all particles, and with transmission electron microscopy (TEM, JEM-2100F; JEOL Inc.). In TEM, the higher vacuum and acceleration voltage leads to enhanced evaporation of volatile matter compared to SEM, thus revealing the underlying shape of the refractory particles only.

## 2.6. Particle chemical composition analyses

The composition of the sub-micron non-refractory particles was measured by the HR-ToF-AMS. The Pika 1.24 toolkit was

used for the analysis of the particle chemical compositions. The Improved-Ambient method was applied for the correction of organic aerosol elemental ratios (Canagaratna et al., 2015). Ionization efficiencies of the HR-ToF-AMS were obtained by measuring 350 nm ammonium nitrate and ammonium sulfate particles. The particle ToF of this AMS was calibrated by 40 to 900 nm dry ammonium nitrate particles. Airflow from the PEAR passed through a HEPA filter was measured for at least 15 minutes during each aging condition to determine the background signals from major gases.



The overall chemical compositions of the samples on QFF were analyzed by two-dimensional gas chromatography coupled with high-resolution time-of-flight mass spectrometry (GC×GC HR-ToF-MS, LECO Pegasus HRT 4D High resolution ToF MS). Comprehensive GC×GC HR-ToF-MS was necessary due to the large number of compounds, diversity of compound classes, and the isomeric complexity of the samples. HR-ToF-MS analysis allowed to determine the exact masses of mass fragments, which were then used for assigning the fragments to the elemental composition classes CH, CHO, or $CHO_2$.

Aromatic ring structures remain predominantly intact during applied fragmentation and the aromaticity of the molecular fragments was assessed based on the double bond equivalents (DBE, $n_C - (n_H/2) + (n_N/2) + 1$), which describes the total amount of double bonds and aromatic rings in the structure. Compounds with DBE $\geq 4$ were classified as aromatics, while DBE $\geq 7$ indicates two-ring aromatic structures.

         Details on GC×GC HR-ToF-MS sample preparation and analysis are available in Supplementary section S1.1.

Besides standard and column bleed, there was no significant elution after pyrene (202 g mol$^{-1}$), and this region was used for background subtraction. The organic matters on the QFFs were very volatile, and likely contained a lot of material adsorbed and condensed from gas phase. Therefore, the relative composition might be affected by the handling of the filter analysis in the GC×GC analyses. Thus, the GC×GC HR-ToF-MS results are considered to provide qualitative information of the chemical composition of the exhaust emissions, rather than exact quantitative composition information.

Thermal-optical carbon analysis (TOCA) was performed using OC-EC Aerosol Analyzer (Sunset Laboratory Inc.) with the IMPROVE A -protocol (Chow et al., 2007). The method separates carbon into EC and OC based on its volatilization during stepwise heating of the filter sample. The OC fraction consists of five bins: OC1-4, with volatility decreasing with the bin number, and OC pyrolyzed during the thermal-optical analysis ('PyrolC'). The protocol also divides EC into three bins, which are, however, presented united due to their marginality compared to OC, and likely overbias due to the high organic

loading on the filter.

**2.7 Determination of particle optical properties**

**2.7.1. Absorption characterization by the aethalometer**

Light absorption by the aerosol was measured online by seven-wavelength aethalometers (AE33, Magee Scientific) based on light attenuation of a PM-laden filter tape. Two instruments were in use, with one continuously monitoring the fresh exhaust

and the second switching between fresh and aged exhausts. A multiple scattering correction factor (C) of 2.39 was used as recommended by Yus-Díez et al. (2021) for the M8060 filter tape in use. Mass-absorption cross section (MAC) of 5.14 m$^2$ g$^{-1}$ was used for the conversion of light attenuation at 880 nm to the equivalent black carbon (eBC) concentrations, as proposed by Elser et al. (2019) for freshly generated aircraft soot.

         A power law function was fit on the aethalometer absorption coefficients at the wavelengths 660 nm, 880 nm, and

950 nm to describe the absorbance by eBC ($b_{abs,BC}$). The absorbance by the brown carbon fraction of the organic aerosol ($b_{abs,}$





$_{BrC}$) was then calculated for wavelengths below 660 nm as the difference in total $b_{abs}$ and $b_{abs,BC}$. Another power law function was then fitted to the $b_{abs,\ BrC}$ to obtain an absorption Ångstrom exponent (AAE; Eq. 1) for the BrC.

$$\frac{b_{abs,BrC}(\lambda_1)}{b_{abs,BrC}(\lambda_2)} = \left(\frac{\lambda_1}{\lambda_2}\right)^{-AAE} \tag{1}$$

Mass absorption efficiency of the organic aerosol ($MAE_{AE,\ BrC}$, in $m^2\ g^{-1}$ of PM) was calculated using the absorbance measured

by the aethalometer for BrC following Eq. (2) for all wavelengths below 660 nm:

$$MAE_{Ae,BrC}(\lambda) = \frac{b_{abs,BrC}(\lambda)}{PM_{SMPS}}, \tag{2}$$

where $PM_{SMPS}$ is the SMPS-based mass estimate with the assumption that all the mass measured by SMPS is organic, which was a sound assumption considering the AMS results on particle composition.

**2.7.2 Absorption measurement using UV-vis spectrophotometry**

The light absorption by the matter collected on the filters was measured directly from the QFFs using a FLAME-T-UV-VIS-ES Spectrometer Assembly, 200-850nm (Ocean Insight, now Ocean Optics), with LPC-250CM Liquid Waveguide Capillary Cell with 250 cm pathlength (WPI). For the analyses, circular spots with 9 mm diameter were punched from the filters. The absorption spectra were determined for untreated filter punches, and from filter punches after extraction with Milli-Q water and after extraction with both Milli-Q water and methanol. All dissolutions were done using 5 ml of liquid (either water or

methanol) using a slow "handshaking/turning" mimicking mixing device for 25 min. The device is not aggressive to ensure that insoluble fraction will not be shaken off the filter.

The absorbance of the water-soluble organic carbon (WSOC) was additionally assessed by extracting the organics from 1.5 cm$^2$ filter punches in 40 mL of ultrapure Milli-Q water. Filter punches were sonicated in a designated ultrasonicator (SONOREX Digitec, Bandelin Inc.) for a total of 30 minutes in three separate 10-minute intervals. The samples were

transferred to a freezer to cool them down in between sonications to minimize any thermal disintegration or chemical transformation of OC during sonication. After the sonication, potential insoluble particles were removed from the extracts by 0.22 µm hydrophilic PTFE syringe filters (Fisherbrand™). The UV-vis absorption spectra by the extract were recorded for the wavelength range of 250 to 700 nm by a spectrophotometer (UV-2401PC, Shimadzu) using aliquots of 3 mL in Quartz cuvettes with 1 cm pathlength. The filter punch used for aqueous extraction was dried under gentle airflow in a clean room for 12 hours.

The concentrations of WSOC in the aqueous extracts were estimated by subtracting the OC measured by the OC-EC analyzer (Lab OC-EC Aerosol Analyzer; Sunset Laboratory Inc) from the dried water-extracted filter punches from the OC measured from the non-extracted filters.

The absorptions (A) on the filters and extracts were determined as in Eq. (3).

$$A = \log_{10}\left(\frac{I_o(\lambda)}{I(\lambda)}\right) \tag{3}$$

using the initial ($I_0$) and measured (I) light intensities. The absorption coefficients for total carbon (TC) and WSOC were determined as in Eqs. (4) and (5), respectively:





$$b_{abs,\ TC}(\lambda) = \frac{A_{filter}}{V_{air}} * A(\lambda) * \ln(10) \tag{4}$$

where $A_{filter}$ is the area of the filter, and $V_{air}$ the volume of air collected on the filter during the sampling, and

$$b_{abs,WSOC}(\lambda) = \frac{A(\lambda)*\ln(10)}{L} \tag{5}$$

where L is the optical length of the cuvette (1.0 cm). The multiplication by ln(10) converts the absorbance given in 10-base logarithm to the natural logarithm.

For the filter-based analyses, the filter loading effects produce a significant source of uncertainty in the absolute values. We applied a multiple scattering correction factor of 3.4 for the filter-based UV-vis $b_{abs}$ values, as suggested by Svensson et al., (2019) for airborne soot collected on QFF. As the material sampled onto filters in this study was quite different 275 from mainly elemental carbon-like soot, the quantitative results can be taken as a rough estimate of the possible ranges in absorption coefficients and MAE values. The same C factor was applied for all extraction steps, introducing uncertainties for comparison of the different extraction steps due to possible differences in the C for the remaining particles. The extraction by water may have caused the remaining material to penetrate deeper to the filter, which may increase the observed absorption. Thus, the $b_{abs}$ values after extraction may be overestimated compared to the original filter.

The mass absorption efficiencies of TC and WSOC ($MAE_{TC}$ and $MAE_{WSOC}$; in units $m^2\ g^{-1}$ of carbon or WSOC) were calculated by Eqs. (6) and (7):

$$MAE_{TC}(\lambda) = \frac{b_{abs,\ TC}(\lambda)}{[TC]} \tag{6}$$

$$MAE_{WSOC}(\lambda) = \frac{b_{abs,WSOC}(\lambda)}{[WSOC]} \tag{7}$$

where [X] is the mass concentration of TC in the air sample collected onto the filter or WSOC concentration in the extract.

**2.8 Radiative forcing modelling**

The radiative forcing efficiency of the aerosols were modelled utilizing a core-shell Mie model similarly as in Leskinen et al. (2023). Wavelength-dependent particle absorption and scattering coefficients and asymmetry parameters were calculated using the N-Mie core-shell model of (Voshchinnikov & Mathis, 1999), which is based on a recursive algorithm by (Wu & Wang, 1991). The complex refractive index of the black carbon core was set to $m_{core} = 1.85 + 0.71$ (Lack & Cappa, 2010). The shell 290 real refractive index was not measured, and two literature values were applied to represent the range of potential values: 1.44, a value published for kerosene (Andrieiev et al., 2020), and 1.55, value published for ambient organic aerosol (Hand & Kreidenweis, 2002). For the imaginary part of the refractive index (*k*), the *k* determined describing the absorbance by the WSOC was calculated by Eq. (8):

$$k_{WSOC}(\lambda) = \frac{MAE_{WSOC}(\lambda) \times \rho \times \lambda}{4\pi} \tag{8}$$

by utilizing the measured effective density $\rho$ of the fresh and aged particles. Similarly, the *k* for total organic aerosol ($k_{BrC,\ Ae}$) was calculated by Eq. (9) using the $MAE_{Ae,\ BrC}$ as a representative of the total organic absorbing fraction, which no longer suffers from excessive filter effects:





$$k_{BrC,Ae}(\lambda) = \frac{MAE_{Ae,BrC}(\lambda) \times \rho \times \lambda}{4\pi} \tag{9}$$

For the exhaust particles, the core volume fraction could be estimated based on the TOCA results as 5 % for the fresh
exhaust particles, and either 1) 0.02 % or 2) 3.9 % for the aged exhaust particles, assuming that 1) EC core remains the same
as in the fresh exhaust, and the increase in TC enhances the shell volume proportionally, or 2) using the values given by the
OC/EC analysis, suffering from misclassification of carbon between the fractions. A selection of core volume fractions
between 0.01 % to 10 % were considered to take into account the uncertainty in the measured OC/EC values and to evaluate
cases where the forming organic aerosol would condense on absorbing soot in ambient air. The core volume fraction was
assumed to be the same for all particle sizes. This is unrealistic, but since shell thickness was not measured, we considered it
a reasonable assumption for the modeling, where several volume fractions were considered, and a mixture of shell thicknesses
would fall between the modeled cases. Finally, the aerosol radiative forcing efficiencies (RFEs), i.e., aerosol forcing per unit
optical depth ($\Delta F \, \tau^{-1}$), were calculated based on the calculated absorption and scattering coefficients and asymmetry parameters
utilizing previously presented equations and constants (Delene & Ogren, 2002; Haywood & Shine, 1995; Sheridan & Ogren,
310   1999).

**2.9 Emission index and SOA yield calculations**

The emission indices (EI) are given as per kg fuel consumed to enable the comparison of our test bench with real engines.
Conversion of dilution corrected concentrations to EIs was done based on the carbon balance between the fuel and exhaust
gases (Eq. 10)

$$EI = [X] * \frac{f_C}{[c_{CO_2}] + [c_{CO}] + [c_{THC}]} \tag{10}$$

where [X] is the mass concentration of given species (in g m$^{-3}$), $f_C$ is the fraction of carbon in the fuel (865 gC kg$_{fuel}^{-1}$), and
$C_{CO2}$, $C_{CO}$, and $C_{THC}$ are the mass concentrations of carbon (gC m$^{-3}$) in the raw gas for the $CO_2$ and CO (measured by FTIR)
and THC (measured by FID). Conversion from volume fractions to mass concentrations was done assuming STP conditions.

Conversion was done separately for each experiment using the averages of respective daily concentrations, as the
gaseous concentrations were very stable within one experimental day. The EI values for the fresh and aged emissions are
presented based for the 4 h runs with DR of 50 as arithmetic mean ± standard deviation during total operation time (in total 12
h or 16 h for fresh and aged exhaust, respectively, or 28 h for the instruments always operated at the fresh exhaust, namely,
FTIR and FID), with the exception of results from the aethalometer and offline results, for which the arithmetic mean ± standard
deviation of samples are given.

Bottom-up estimation of the SOA mass formation was conducted based on the VOCs measured by PTR-ToF-MS,
similarly as in Hartikainen et al. (2024). The production of SOA was estimated based on previously published yields (Y) for
individual VOCs in NO$_x$-limited conditions by Eq. (11):

$$SOA_{estimate} = \sum_i Y_i * \Delta[VOC]_i \tag{11}$$



where the $\Delta[VOC]_i$ represents the difference in concentration with or without photochemical processing for a decaying

compound. This approach accounts for the traditional SOA precursors, including aromatic hydrocarbons and oxygenated aromatics, and gives lower limits on the contributions of single compounds or compound groups to the observed SOA. The applied yields are available in Table S4 for each ion. The resulting estimate of SOA was compared to the observed SOA formation to gauge how well the observed VOCs explain the actual secondary aerosol formation.

## 3 Results and discussion

### 3.1 Gaseous emissions

#### 3.1.1 Gases in the fresh exhaust

The EIs for CO, $NO_x$ and THC in the fresh exhaust emissions at the selected operating point were $92 \pm 2.7$ g $kg_{fuel}^{-1}$, $2.0 \pm 0.07$ g $NO_2$ $kg_{fuel}^{-1}$, and $38 \pm 3.6$ gC $kg_{fuel}^{-1}$, respectively. $SO_2$ or $NH_3$ concentrations were below instrument detection limit. The total VOC EI measured by the PTR-ToF-MS in the m/z range of 40 to 200 was $44 \pm 5.5$ g $kg_{fuel}^{-1}$, or $32 \pm 3.9$ gC $kg_{fuel}^{-1}$

expressed in carbon mass basis. Overall, the gaseous EIs in the fresh exhaust emission were comparable to those reported from the taxi/ground idle mode of older generation gas fan engines (International Civil Aviation Organization, 2022). Older engines also do not have high pressure, whereas in newer engines the gaseous emissions are typically reduced by the higher compression and higher combustion efficiency. The pressure of newer engines is also lower in idling and taxiing conditions, and thus organic gaseous emissions of jet engines are generally the highest at low loads (Cross et al., 2013; Kilic et al., 2017;

Presto et al., 2011). At higher loads the VOC emissions are typically reduced due to the enhanced combustion efficiency (Beyersdorf et al., 2012; Cross et al., 2013; Kilic et al., 2017; Rohkamp et al., 2024).

The PTR-ToF-MS spectra of the fresh exhaust gases included aliphatic hydrocarbons or hydrocarbon fragments, oxygenated (carbonylic) aliphatic compounds, and aromatic hydrocarbons and oxygenated aromatics. EIs of individual compounds are illustrated in Fig. 3a and available in Table S4. Aromatic hydrocarbons and oxygenated aromatic compounds

contributed to 19 % and 13 % of the fresh spectra measured by PTR-ToF-MS, respectively. The most prominent aromatic hydrocarbon ions were $C_{11}H_{16}$-$H^+$ ($0.68 \pm 0.07$ g $kg_{fuel}^{-1}$), $C_{12}H_{18}$-$H^+$ ($0.64 \pm 0.06$ g $kg_{fuel}^{-1}$). Aromatic compounds measured by the PTR-ToF-MS included the smallest polycyclic aromatic hydrocarbons (PAHs), namely, naphthalene ($C_{10}H_8$-$H^+$; $128 \pm 23$ mg $kg_{fuel}^{-1}$) and methylnaphtalenes ($C_{11}H_{10}$-$H^+$, $343 \pm 48$ mg $kg_{fuel}^{-1}$). The overall contribution of aromatic hydrocarbons to the total VOC EI was similar to those previously measured for large gas turbine -engines on low ($\leq 7$ %) loads, but the ratios

of substituted aromatics to benzene ($C_6H_6$-$H^+$, EI $258 \pm 18$ mg $kg_{fuel}^{-1}$) were here higher than in previous observations where benzene has been the dominant aromatic compound in the gas phase (Beyersdorf et al., 2012; Kilic et al., 2017; Presto et al., 2011).

Individual aromatic compounds were also quantified offline, with the highest EIs observed for substituted naphthalenes ($87 \pm 17$ mg $kg_{fuel}^{-1}$ for 1,2-dimethylnaphthalene, $84 \pm 14$ and $143 \pm 24$ mg $kg_{fuel}^{-1}$ for 1- and 2-methylnaphthalene,




respectively; Fig. 4), with a reasonable agreement with the PTR-ToF-MS-based EIs. The organic gases included considerable amounts of aliphatic hydrocarbons that were not observable by PTR-ToF-MS. The concentrations of alkanes measured offline by GC-MS exceeded the calibration limits by a factor of 8 to 14 (Table S5), suggesting that their EIs were in the range of hundreds of mg $kg_{fuel}^{-1}$ for individual compounds. The highest concentrations were measured for $C_{11}$ to $C_{14}$ alkanes, followed by $C_{15}$ and $C_{16}$ alkanes. Previously, unburned fuel has been noted as a dominant source of n-alkanes and other intermediately-

or semi-volatile organic compounds from idling aircrafts, but they can also arise from fragmentation of larger compounds (Cross et al., 2013; Presto et al., 2011).

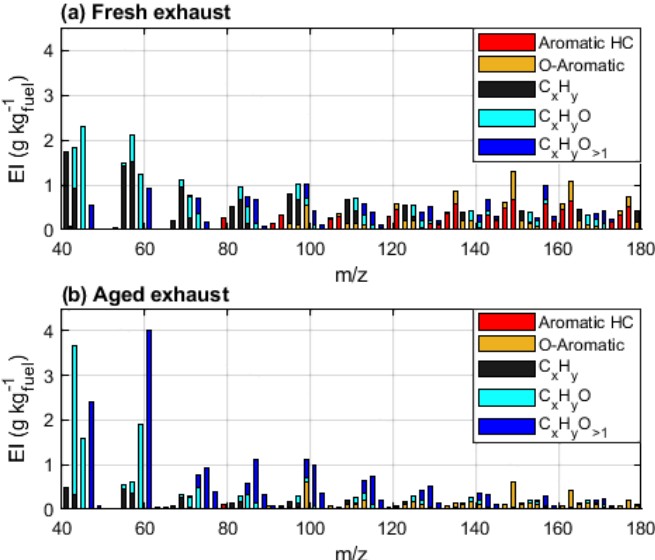

**Figure 3.** Average VOC EIs measured by the PTR-ToF-MS in fresh (a) and photochemically aged (b; DR50) exhaust gases. The high-resolution peaks are presented stacked in unit mass resolution for visual clarity.

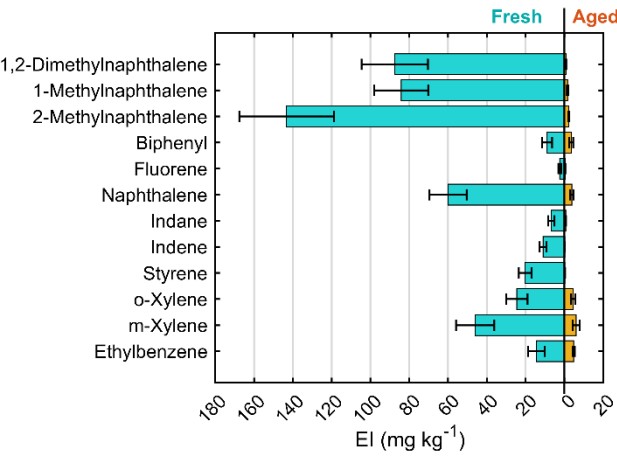


**Figure 4.** Emission indices of aromatic compounds analysed offline by GC-MS. Error bars indicate the standard deviations between replicates (n = 6 or 3 for fresh or aged exhaust, respectively).



### 3.1.2 Gas-phase changes by the photochemical processing

The PTR-derived total VOC EI after photochemical processing was $34 \pm 5.2$ g $kg_{fuel}^{-1}$ (or $20 \pm 3.2$ gC $kg_{fuel}^{-1}$). Photochemical
aging reduced especially the concentrations of aromatic hydrocarbons measured by the PTR-ToF-MS, which were 93 % lower
after 2 eqv. d of photochemical aging than in fresh exhaust (Fig. 3b, Table S4). PAHs measured by the PTR-ToF-MS sustained
only 1 % of the fresh concentrations after 2 eqv.d. The decrease of individual PAHs was confirmed by the offline GC-MS
results, with a 66 % to 99 % lower EIs after photochemical processing (Fig. 4). The decay in the aromatic compounds was
generally in line with the reaction rates of the individual compounds with OH·. For example, the substituted naphthalenes were
completely (98 % to 99 %) consumed during the photochemical processing, whereas the less reactive ethylbenzene or biphenyl
decayed only by 67 or 60 %. The offline-measured n-alkanes also decayed significantly, with 50% to 80 % lower signals after
the photochemical processing (Table S5).

Photochemical processing formed notable amounts of small carbonylic compounds, such as m/z's 43.02 ($C_2H_2O$-
H+), 59.05 ($C_3H_6O$-H+), and 63.01 ($CH_2O_3$-H+) (Fig. 3b). These compounds are typical fragments from oxidizing gaseous
and particulate organic aerosols in combustion exhaust emissions (Hartikainen et al., 2020; Hunter et al., 2014; Ziemann &
Atkinson, 2012). Simultaneously, the PTR-based EIs of oxygenated PAHs was enhanced by a factor of 1.4 by addition of
series of highly oxygenated VOCs with even three or four oxygens and DBE over 4 (Fig. 3b, Table S4), indicating formation
of oxidized aromatic structures or cycloalkanes with unsaturated substitutes (Hunter et al., 2014; Pagonis & Ziemann, 2018).

### 3.2. Physical particle properties

### 3.2.1 Density and morphology

Density of both fresh and aged particles increased with increasing particle size (Fig. S2), indicating spherical particle
morphology and potential instrument artifacts in the lowest assessed sizes. (Durdina et al., 2014; Saffaripour et al., 2020)SEM
and TEM micrographs confirm the shape of both fresh and aged particles as generally spherical, and fractal-like aggregate
structures were markedly scarce (Fig. S3). The few non-volatile particles observed in TEM had distinctly closed aggregate
structure (Fig. S4). The observed sphericality contrasts with previous assessments of effective density of aircraft exhaust
particles, where a power-law relationship typical for fractal soot aggregates has been found between mobility size and $\rho_{eff}$ of
aircraft particles (Durdina et al., 2014; Saffaripour et al., 2020).

The size-resolved density of particles decreased by photochemical aging likely due to compositional differences; see
Section 3.3. However, the growth in average particle size during the aging processes increased the average particle density.
The average effective density obtained by the AMS-SMPS method for the aged exhaust particles ($1.35 \pm 0.03$ g cm$^{-3}$) agrees
well with those of the AAC-SMPS for the average aerodynamic size of 80 nm ($1.37 \pm 0.04$ g cm$^{-3}$), whereas the AAC-SMPS-
based density of the fresh 80 nm particles was $1.52 \pm 0.06$ g cm$^{-3}$. For the calculation of PM$_{SMPS}$ and $k$, density of 1.35 g cm$^{-3}$
was applied for both the fresh and aged particles of all sizes.



### 3.2.2 Particle number, mass, size, and LDSA

The PM mass and number EIs in the fresh exhaust emission were $0.06 \pm 0.01$ g kg$_{fuel}^{-1}$ and $(5.3 \pm 1.5) \times 10^{15}$ # kg$_{fuel}^{-1}$, based on the SMPS measurements. TEOM-based mass (PM$_{TEOM}$) EI was $0.06 \pm 0.02$ g kg$_{fuel}^{-1}$, agreeing closely with the PM$_{SMPS}$, indicating that the sampled particles were not very volatile. The fresh particulate emission indices are comparable to in-use aircraft engines (International Civil Aviation Organization, 2022; Mazaheri et al., 2009; Presto et al., 2011). The particle size distributions (PSDs) of the fresh samples were bimodal, with modes at 13 nm and 25 nm (Fig. 5), and potentially influenced

by the sampling and dilution settings, as the cooling exhaust emission had the potential to coalescence prior to the SMPS measurement. The small particle sizes agree with previously measured sizes of aircraft exhaust particles although those are most often presented for the non-volatile fraction only (Corbin et al., 2022; Saffaripour et al., 2020). The evaporation sticks inside of the combustion chamber atomize the liquid kerosene much more finely than in older engines, where direct injection was achieved via pressure nozzles. The formation of smaller droplets allows enhanced mixing with combustion air, and

consequently more complete combustion of the kerosene. This also allowed the formation of ultrafine particles, which is characteristic of newer engines.

Photochemical exposure led to extensive new particle mass formation, with secondary EIs of $18.5 \pm 1.9$ g kg$_{fuel}^{-1}$ and $8.7 \pm 0.77$ g kg$_{fuel}^{-1}$ for PM$_{SMPS}$ and PM$_{TEOM}$, respectively, at DR50 experiments, with respective enhancement ratios of 290 and 130 compared to the fresh EIs. The disagreement between the secondary PM$_{SMPS}$ and PM$_{TEOM}$ indicated that in contrast to

the fresh exhaust particles, much of the aged particulate matter is semivolatile. The observed SOA formation potential was notably higher than previously observed for exhaust emissions of full turbine engines operated on low loads, both in absolute EI and in enhancement ratio (Kilic et al. 2018, Miracolo et al. 2011), which is in line with the relatively high precursor gas emissions.

Photochemical processing increased the number EIs to $(2.6 \pm 2.0) \times 10^{15}$ kg$_{fuel}^{-1}$ in the DR50 experiments, while

particle geometric mean diameter (GMD) increased to $78 \pm 2.8$ nm with a unimodal PSD. Higher dilution decreased particulate condensation sinks, and therefore enhanced new particle formation via nucleation. As a consequence, the particle number was considerably enhanced, while average particle size was correspondingly smaller compared to the experiments at lower dilution, e.g., PN of $7.4 \times 10^6$ # kg$_{fuel}^{-1}$ and GMD of 50 nm at 1.7 eqv.d (Fig. 5). The formed particle mass was, however, only slightly higher for the DR50 experiments (Fig. 6a). Changes in particle number or size due to coagulation in the PEAR were estimated

to be negligible for both dilutions.



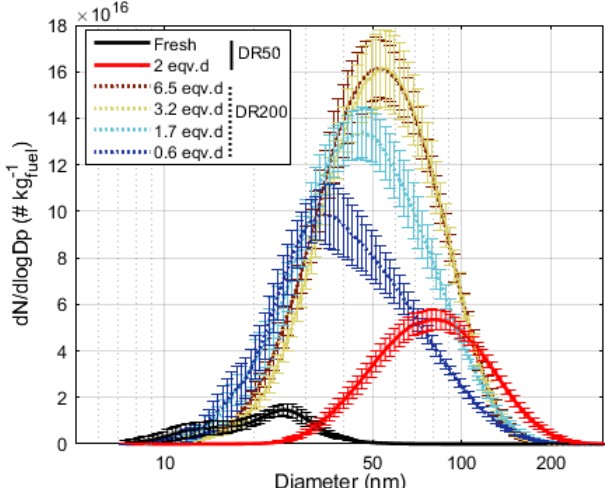

**Figure 5. Particle number size distributions measured by the SMPS for the fresh and aged exhaust emissons. Error bars denote the standard deviations over the whole experimental time (12 or 16 h for the DR50 fresh and aged experiments, respectively; 40 to 70 min for the DR200 experiments).**

The EIs for alveolar LDSA were $370 \pm 86$ cm$^2$ kg$_{fuel}^{-1}$ and $8900 \pm 460$ cm$^2$ kg$_{fuel}^{-1}$ for the fresh and aged exhaust emissions, respectively. The enhancement ratio of LDSA was 24, which is distinctly below the enhancement in total particulate mass. This is due to the size dependency of lung deposition efficiency: in the alveolar region, the deposition efficiency peaks at 50 % at 20 nm, and is below 20 % for particles above 50 nm in mobility diameter (Hinds & Zhu, 2022). This means that for the size range of the fresh exhaust particles, any particle growth would decrease their deposition efficiency in the lungs and

head airways.





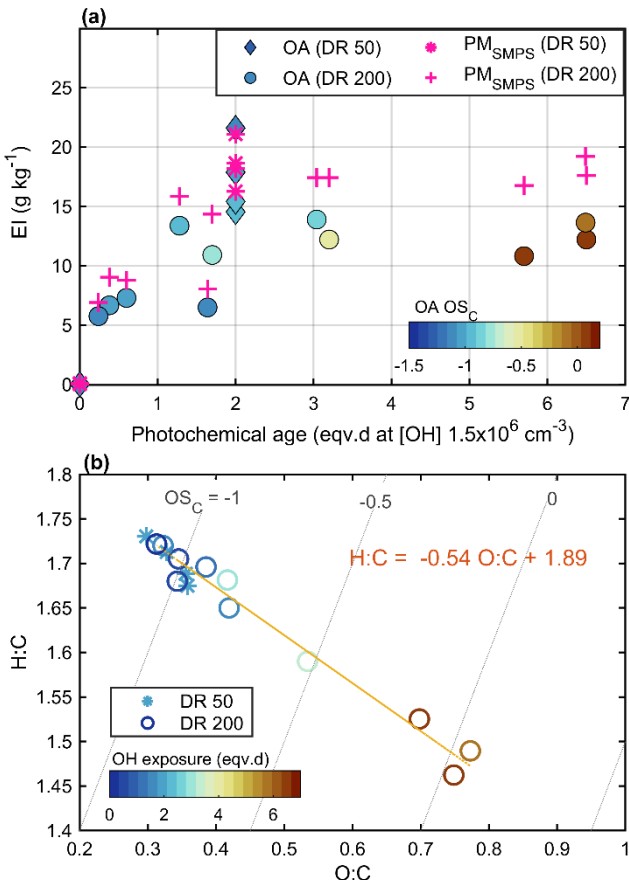

**Figure 6. Mass emission indices for the particles measured by SMPS (PM$_{SMPS}$) and organic aerosol (OA) measured by the AMS, with average carbon oxidation state indicated by color (a), and average elemental ratios of the organic aerosol, with OH exposure indicated by color (b).**

## 3.3 Particle chemical composition

### 3.3.1 Fresh particle composition

The density measurements as well as the agreement with PM$_{SMPS}$ and PM$_{TEOM}$ suggest that the fresh exhaust particles were not particularly volatile. The fresh filter samples, in contrast, included a large share of semi- to low-volatile organic species adsorbed from gas phase onto the QFF (Fig. 1b). This increased the filter-based carbon concentrations especially for the primary exhaust samples, and led also to the discrepancies between the online- and offline absorption measurements. The OC:EC ratio, determined by TOCA, was 23 for the fresh exhaust samples, while the total carbon EI was $0.29 \pm 0.09$ g kg$_{fuel}^{-1}$. The OC:EC ratio was higher than previously observed in low thrusts (Elser et al., 2019; Miracolo et al., 2012; Presto et al., 2011), assumably mainly due to the lack of back-up filter for gas-phase adsorption correction. The size of fresh particles was below the optimal AMS measurement range, and their concentrations were too low for obtaining online compositional information by the AMS.





Little aliphatic hydrocarbons and only a low extent of one- and two-ring aromatic structures were detected in the fresh exhaust samples (Fig. 1b). Noticeably, the homologous series of aliphatic hydrocarbons in the fuel was converted to a series of aliphatic carbonyls, mostly ketones, in the fresh exhaust emission. Further, a prominent high-intensity, non-resolvable hump was observed in the center of the contour plot. Single peaks in this hump and the m/z values of the summed mass spectra

indicate a variety of incomplete combustion products, like carbonyls and alcohols. Few higher PAHs were found in very low intensities, such as pyrene, fluoranthene and multi-alkylated phenanthrene.

### 3.3.2 Particle composition after photochemical aging

The photochemically aged particles were almost purely organic matter. Based on the AMS results, the aged EIs were $17.4 \pm 2.9$ and $0.3 \pm 0.2$ g $kg_{fuel}^{-1}$ for OA and $SO_4$, respectively. $PM_{SMPS}$ and AMS OA agreed for the DR50 experiments, whereas

there was a 30 % disagreement for the DR200 experiments (Fig. 6a). Most of this discrepancy is due to the small size of the particles, as approx. 20 % of particle mass fell under the limit of sufficient collection efficiency at the higher dilution experiments. TEM imaging revealed the secondary particles to contain both volatile material and relatively stable, even refractory organics that remained visible under the vacuum and only partly evaporated when focusing the electron beam (Fig. S4).

The photochemical processing increased the total carbon EI measured from the QFF to $8.5 \pm 1.4$ gC $kg_{fuel}^{-1}$, based on TOCA. This agrees with the AMS-based OC EI, which was 11 gC $kg_{fuel}^{-1}$ at the measured OM:OC ratio of $1.6 \pm 0.03$. The enhancement ratio in total carbon content measured by TOCA was 30, which is relatively low compared to the SMPS based estimate due to the excessive gas-phase adsorption especially in the primary exhaust sampling. Of the aged OC, a relatively high share ($34 \pm 4$ %) was charred during the heating steps of TOCA ('pyrolytic C'; Fig. S5). This is typical for high-molecular

weight organic molecules (Chow et al., 2007), which here can be expected to have formed during photochemical oxidation and nucleated into particles in the PEAR from the relatively particle-free initial exhaust emissions. The fraction of OC3 to total carbon was notably lower ($20 \pm 1$ %) after aging than in the fresh exhaust emissions ($36 \pm 10$ %), but the absolute concentrations were significantly higher for all the fractions after aging. In TOCA, also EC was observed to be enhanced by a ratio of 21; however, this additional EC is suspected to be either misclassified refractory OC, or to arise artificially from the OC charred

during the sample heating, as no new mature soot is to be formed in the photochemical oxidation of organic carbon.

GC×GC-MS results revealed an increase in the chemical diversity of the organic matter, confirming that the photochemical processing of the emissions led to formation of a range of new compounds. The relative amount of oxygen and the number of both CHO and CHO2 fragments increased after aging (Fig. 1c). However, the qualitative fragmentation patterns of oxidized species, indicating compounds functionality, were comparable to the those detected from the fresh filter. Higher

number of aromatic species were observed in the aged exhaust samples compared to the fresh exhaust samples, as indicated by the increase in both one-ring (DBE 4–7) and two-ring (DBE ≥ 7) molecular fragments. This was consistent with the decay of gaseous aromatic compounds, photo-oxidation of which can be expected to result in particle-phase products with aromatic



structures. In addition to the increase in boiling point by the photo-oxidation leading to condensation, the presence of substantially higher amounts of particles enhanced partitioning of semivolatiles onto the particle phase.

The O:C and H:C -ratios of the aged OA were $0.33 \pm 0.03$ and $1.70 \pm 0.02$ for the DR50 experiments, as measured by the AMS. The particles aged at DR of 200 had slightly higher average organic carbon oxidation state ($OS_C$) than those aged at lower dilution at similar photochemical exposure (Fig. 6b). This is explained by the requirement for more gas-phase oxidation steps prior to the new particle formation via nucleation, compared to those required for condensation when more particles are available as condensation sinks (Table S3; Czech et al., 2024). In addition, the higher particle concentration in

less diluted conditions promoted partitioning of less oxidized semivolatiles into particle phase. Some of the differences between dilutions may also arise from the lack of collection of smallest particles by the AMS in the higher dilution experiments, in case there were differences in composition of particles of different sizes. The differences in the aging conditions, such as the in variation in the ratio of photolysis to OH· exposure (Table S3) may also have caused differences to the particle formation pathways, although the linearity of the slope suggests that reaction pathways remained similar between experiments.

$OS_C$ increased as a linear function of photochemical age (Fig. 6b), with a slope of -0.54 for H:C/O:C (orthogonal linear regression). This a typical slope for photochemical aging of urban organic aerosol and representative for, for example, addition of both acid and alcohol functional groups without fragmentation or the addition of acid groups with fragmentation (Chen et al., 2015; Ng et al., 2011). Based on the gas phase analyses, fragmentation can be expected to have occurred in significant amounts, and the formation of acidic groups is implied also by the GC×GC-MS results by the formation of new

CHO2 fragments (Fig. 1c).

### 3.4 SOA precursors in the exhaust emissions

The PM masses appeared to have reached their peak within roughly 2 eqv.d and mass increase with extended photochemical exposure was marginal. This indicates that the available precursors were oxidized within relatively short exposure timescales, which agrees with the OH· reactivities of the assumed main SOA precursor species, including both aromatic compounds and

alkanes. SOA formation depends on the available seed concentration (Srivastava et al., 2022), explaining the slightly higher SOA-mass EI for the DR50 conditions (Fig. 6a). Ratio of $THC_{FID}$ to $NO_x$ was 70 ppmC ppm$^{-1}$ in the fresh exhaust emission, so $NO_x$-restricted yields were applied in the bottom-up estimation. The SOA estimated to arise from the decay in compounds measured by PTR-ToF-MS was 5.0 g $kg_{fuel}^{-1}$, which covers 27 % of the observed total secondary $EI_{SMPS}$. $C_{10}$-, $C_{11}$-, and $C_{12}$-aromatic hydrocarbons were the most important SOA precursors of the compounds included in this bottom-up estimation (Fig.

S6), forming 14 %, 14 %, and 12 % of the total estimated SOA, respectively. The most prominent single ions were $C_{10}H_{14}$-H+ (m/z 135.11), $C_{11}H_{16}$-H+ (m/z 149.13) and $C_{12}H_{18}$-H+ (m/z 163.14), with 7.8 %, 8.1 %, and 7.0 % individual shares to the total estimated SOA. The limited availability of $NO_x$ for the subsequent reactions of the peroxyradicals formed by OH· oxidation likely enhanced the SOA formation from aromatic precursors compared to $NO_x$-rich conditions (Srivastava et al., 2022).

       The n-alkanes measured by GC-MS were estimated to contribute to secondary particle formation by minimum of 1

g of SOA $kg_{fuel}^{-1}$, assuming previously published high-$NO_x$ yields (Lim & Ziemann, 2009). The restricted $NO_x$ availability



likely also favoured SOA formation compared to higher NO$_x$ availability (Hunter et al., 2014; Loza et al., 2014), but the gap between the observed and estimated SOA remains ubiquitous. In addition to the n-alkanes, cycloalkanes are suspected to have participated in the SOA formation, as the GC×GC-MS analyses revealed the formation of alkylated cyclohexanones and - pentatones as well as alkylated tetrahydropyranones (NIST library spectrum similarity >70 %) during the photochemical

processing. Relatively high SOA yields have been reported for cycloalkanes compared to linear alkanes, especially at low-NO$_x$ conditions, because the cyclic structure protects the molecules from fragmentation during oxidation (Hunter et al., 2014; Lim & Ziemann, 2009). The absorbing properties, discussed in the next section, and the addition of highly oxidized organics visible in the higher mass range of the PTR-ToF-MS spectra further support the formation of secondary organics from aromatic or cyclic structures, with products largely partitioning to the particle phase (Hunter et al., 2014; Pagonis & Ziemann, 2018).

Cycloalkanes can be key ingredients in jet fuels due to their advantageous fuel properties, such as high energy density and low freezing point, and the prevalence of cycloalkanes in aircraft exhaust gases and their secondary products and SOA yields should be considered in future studies.

**3.5 Light absorbance**

In the fresh exhaust emissions, the absorption measured by aethalometer at 880 nm corresponded to eBC EI of 4.3 ± 0.5 mg

kg$_{fuel}$$^{-1}$. For the aged exhaust emissions, aethalometer-based eBC EI increased to 90 ± 16 mg kg$_{fuel}$$^{-1}$. Similarly, the offline UV-vis spectrophotometry results from filter-based spectrophotometry analyses support the enhancement of absorption in the near-IR range (Fig. 7a-b). Both methods may suffer from loading effects, which were likely enhanced in the more loaded aged samples compared to the fresh exhaust samples due to the increased scattering and shadowing effects. Photochemical aging led to excessive particle formation, and the absorbance by BrC$_{Ae}$ was manifold for the photochemically aged exhaust particles

(Fig. 7c). Simultaneously, the AAE$_{BrC}$ fit to all wavelengths from 370 to 590 nm rose from 7.1 ± 0.3 to 10.7 ± 0.9 by the photochemical aging. Absorption as well as its wavelength-dependency depended on the extent of photochemical aging, with increasing OH· exposure enhancing the absorption (Fig. 7d).

The filter-based absorbance measurements indicated that the water-insoluble fraction may have constituted half of the total absorbance potential for both the fresh and aged exhaust samples, with the rest accounted for by either methanol-

soluble (but water-insoluble) organics or methanol-insoluble particles, including the potential elemental carbon (Fig. 7a-b). Photochemical processing decreased the absorption by the methanol-insoluble fraction by 60 % at 550 nm, indicating a decay in of very high-molecular weight compounds.

The b$_{abs, BrC}$ measured by the aethalometer contained both water soluble and water-insoluble fractions of the absorbing organic matter, whereas the QFF-based b$_{abs, WSOC}$ was enhanced by the gas phase adsorption. As a result, the absorbance

measured by aethalometer for the BrC-fraction ended up being remarkably well in line with the UV-vis spectrophotometer analyses for the fresh WSOC (Fig. 7c). In the aged exhaust emission, the aethalometer-based b$_{abs, BrC}$ was roughly double the b$_{abs,WSOC}$. This discrepancy is in line with the contribution of water-insoluble absorbing matter in the filter-based UV-vis



analyses. There, the relative impact of gas-phase adsorption on the QFF was lower in the aged than for the fresh exhaust samples.

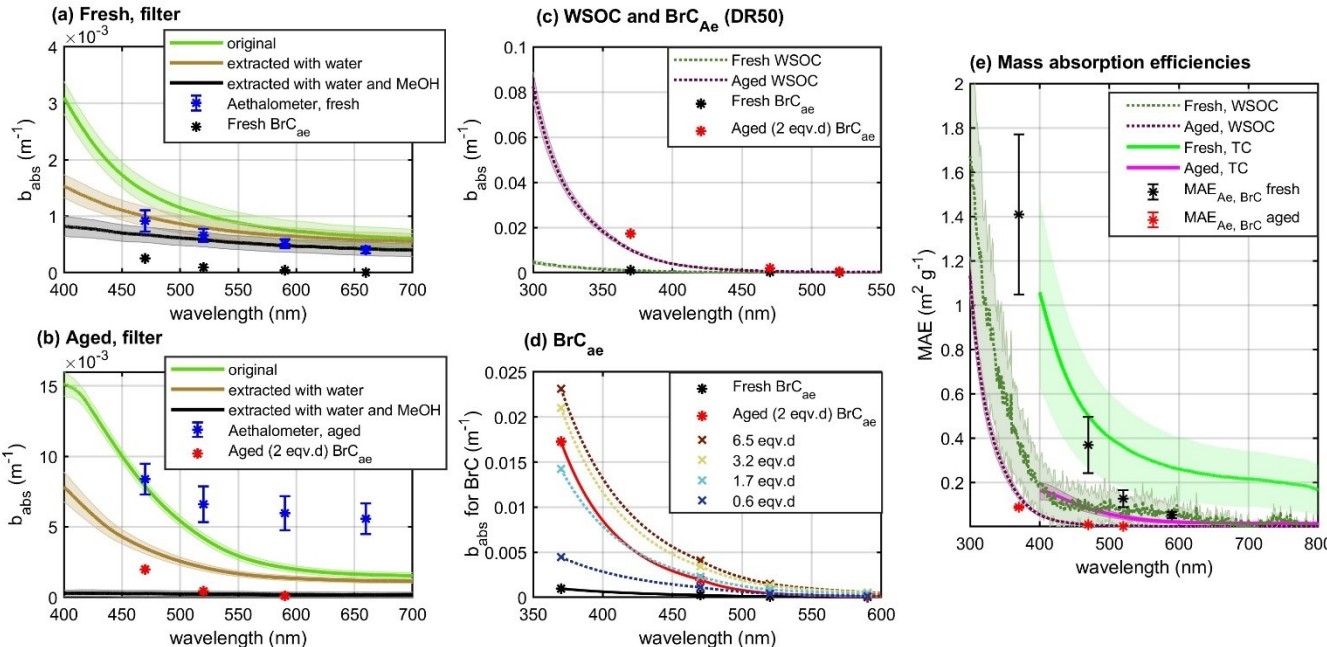


**Figure 7. Absorption coefficients for fresh (a) and aged (b) exhaust emissions measured by the aethalometer and filter-based UV-vis spectrophotometry at the different extraction stages; absorption coefficients for absorption by WSOC (c) and aethalometer-based brown carbon for a range of photochemical ages (d); and mass absorption efficiencies for total carbon content (MAE$_{TC}$) and for water-soluble organic carbon fraction (MAE$_{WSOC}$) (e). Absorption coefficients are presented at dilution-corrected raw exhaust**
**concentration.**

Both fresh and aged WSOCs had evident absorbing characters towards the lower visible-to-UV-wavelength range. For the total organic PM, MAE$_{Ae, BrC}$ estimated based on aethalometer b$_{abs, BrC}$ and PM$_{SMPS}$ were $1.4 \pm 0.4$ m$^2$ g$^{-1}$ and $0.09 \pm 0.003$ m$^2$ g$^{-1}$ at 370 nm, and $0.13 \pm 0.04$ m$^2$ g$^{-1}$ and $0.002 \pm 0.0002$ m$^2$ g$^{-1}$ at 520 nm for fresh and aged exhaust emission, respectively (Fig. 7e). The MAEs per gram of carbon would be lower proportionally to the OC:OM ratio (1.6 for aged exhaust).
For the WSOC sampled to the QFF, MAE$_{WSOC}$ of $0.54 \pm 0.25$ m$^2$ gC$^{-1}$ and $0.17 \pm 0.02$ m$^2$ gC$^{-1}$ at 365 nm were determined for fresh and aged WSOC, respectively (Fig. 7c). These MAE$_{WSOC}$ are at least an order of magnitude higher than previously reported for photochemical SOA formed from alkanes in the absence of NO$_x$ (below 0.01 at 405 nm; Lambe et al., 2013; Li et al., 2021; Updyke et al., 2012). SOA formed from aromatic hydrocarbons, on the other hand, have often been noted to contain chromophores and to have substantially higher MAEs than the SOA observed here (Lambe et al., 2013; Moise et al., 2015;
Nakayama et al., 2013). This further confirms the formation of SOA from a mixture of precursor classes.





## 3.6 Direct radiative forcing efficiency

The imaginary refractive indices for the fresh organic exhaust particles ($k_{BrC,Ae}$) were 0.056 and 0.0071 at 370 and 520 nm, respectively. For the WSOC, $k_{WSOC,}$ were 0.021 and 0.0070 at 365 and 550nm, respectively. Aging decreased the aethalometer-based total $k_{BrC,Ae}$ to 0.0035 (at 370 nm) and 0.00013 (at 520 nm), and $k_{WSOC,}$ 0.066 (at 365 nm) and 0.00059 (at 550 nm). Such
particulate matter can be classified as 'weakly' (fresh) or 'very weakly' (aged) absorbing brown carbon (Saleh, 2020). The direct radiative forcing efficiency of the particulate matter was estimated based on a Mie model, where the exhaust particles were assumed to be internally mixed and consist of core-shell like brown carbon -particles with a minor EC core. Namely, EC core fractions of 0.01 % to 10 % were used based on the TOCA results, where the OC:EC ratio of the fresh exhaust samples (21) would correspond to EC volume fraction of 5 %, and the fresh EC to 0.02 % core to aged total carbon.

The Mie-model results indicate that the fresh exhaust particles would have relatively efficient warming effect on climate, but photochemical aging would turn the effect into clearly cooling (Fig. 8). This shift to negative radiative forcing effect is due to the enhancement of scattering, caused by 1) addition of assumably effectively scattering secondary organic matter, and 2) growth in size to a more efficiently scattering size range, which can be observed in Fig. 8 by comparing the cases modelled with a non-absorbing coating. For the fresh aerosol, addition of a weak absorbing character to the coating in
the model appears to slightly increase the RFE compared to the assumption of completely non-absorbing shell, when the core-fraction is minor. For the aged aerosol, slight absorbing character increases the RFE at 370 nm by 10 – 20 %, but it remains cooling in all modelled cases. The impact of the formed very weakly absorbing coating on the total RFE appears, however, minor, as the differences in RFEs compared to non-absorbing coating -assumption were below 2 % for the wavelengths above 370 nm.



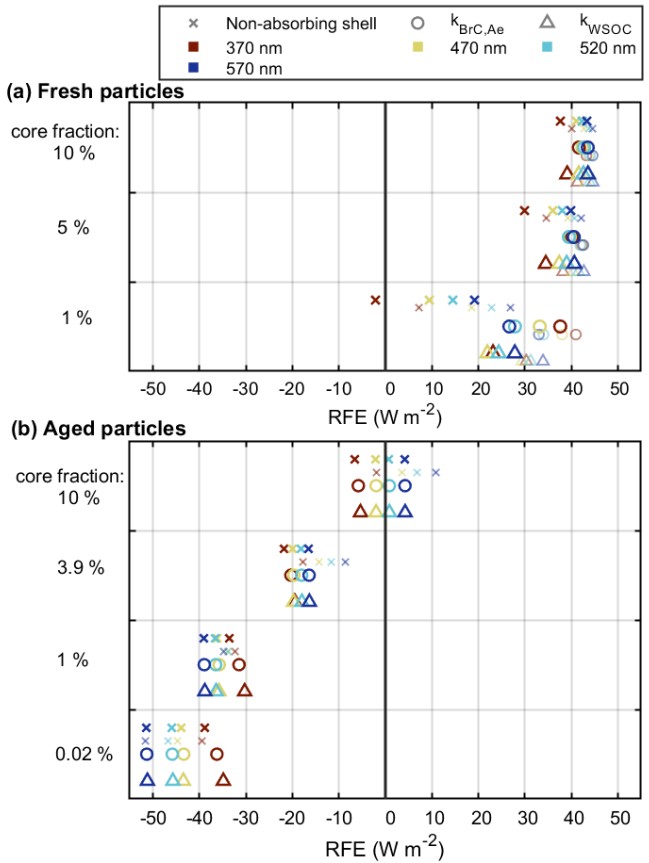

**Figure 8. Radiative forcing efficiencies (RFE) of the fresh (a) and aged (b) particles for the four lowest aethalometer wavelengths, either assuming a non-absorbing shell or utilizing the imaginary refractive indices measured by aethalometer for brown carbon ($k_{BrC,Ae}$) or measured for WSOC ($k_{WSOC}$). Values obtained using the real RI of 1.55 are shown in larger markers, while the smaller and lighter markers indicate RI = 1.44.**

## 4 Conclusions

The physical and chemical characteristics of exhaust emissions from a small-scale jet engine burner running on kerosene-based jet fuel were transformed by photochemical processing in an oxidation flow reactor. Fresh exhaust emissions contained relatively high amounts of organic matter, which was mainly partitioned to the gaseous phase and contained a major SOA formation potential. Photochemical oxidation was estimated to lead to a factor of 300 increase in particulate mass. $C_{10-12}$ - aromatic hydrocarbons were the most prominent of the measured SOA precursors, but most of the new particle mass formation couldn't be explained by the oxidation of the volatile gaseous precursors measurable by PTR-ToF-MS or the n-alkanes assessed separately by GC-MS. Of the unquantified precursors, cyclic alkanes were hypothesized to have contributed significantly to the SOA formation based on the molecular structures observed in the aged exhaust aerosols. Jet fuels may contain higher fractions of cycloalkanes than e.g., gasoline or diesel fuels to meet the requirements for higher energy density, lower freezing





point, and suitable lubricant properties, and the contribution to cycloalkanes to SOA from aviation should be constrained in the future. As the precursor compounds are relatively quick to react in photochemical conditions, the secondary formation can be expected to occur within the timescale of hours to days.

The particle size also increased drastically upon secondary aerosol formation. Shifts in size distributions are to be expected also in the real atmosphere, although the sensitivity of growth in size to the extent of dilution highlights the importance of background particle concentration to new particle formation. Nevertheless, the aged particles remained in the UFP range (i.e., below 100 nm), which is the typical size range of particles originating from airports resolved from urban air (Riley et al., 2021; Stacey, 2019). This indicates that the aviation-derived ambient ultrafine particles can originate also from photochemical processing of originally gaseous organic compounds, and that secondary organic aerosol from aircraft exhaust emissions can be a notable source of organic particulate matter, including UFPs, to urban air.

As the amount and composition of the exhaust particles changed with the degree of atmospheric processing they underwent, the implications of long-transported aircraft exhaust emissions to air quality will depend on their 'atmospheric age' and the composition of the ambient air. The observed secondary formation of gas-phase carbonyls, transformation in the polycyclic aromatic content, and the increased oxidation state of the organic particles can also alter the toxicity of the exhaust emissions. This work focused on mimicking the airport operation by replicating the average emissions of an LTO cycle, and the reported characteristics are representative of on- or near-ground exhaust emissions originating from an airport. Further, the presented characteristics are for secondary aerosol from kerosene-combustion exhaust emission purely, as the experiments were conducted without lubrication oil. Lubrication oil has been noted to be a notable precursor for soot or volatile particulate matter formation from full scale turbine engines (Fushimi et al., 2019; Ungeheuer et al., 2022; Yu et al., 2010), and the impacts of the mixing of lubrication oil and kerosene combustion exhausts on the aging products and impacts from aircrafts on a larger range of operation conditions should be thoroughly studied in the future.

The fresh particles were estimated to have positive radiative efficiency in the atmosphere, i.e., climate warming effect. Photochemical aging led to prominent formation of very weakly absorbing organic carbon, which enhanced the absorption but was ultimately estimated to shift the particles into effectively cooling due to the simultaneous increase in scattering. Failing to consider the secondary formation would, in other words, result in mis-estimation of the direct radiative forcing caused by the exhaust particles. Here, we focused on the direct radiative forcing effect caused by the particles near airport, where they may cause local warming or cooling effect while contributing to the formation of near-airport haze. The transformation pathways for the higher engine power exhaust emissions from in-air operation may be inherently different, and the changes in emission contents by atmospheric aging processes would influence cloud and contrail formation more than airport operations.

The volatile particulate matter emitted either directly from the engine or, as showcased here, formed in large quantities in the air as secondary particulate matter, may dominate the particulate pollutants from aircrafts, especially for the on-ground, low-power operation. Aviation emissions are currently under considerable transformation, as novel engine designs and new sustainable aviation fuels (SAFs) are introduced with the aim of less polluting and less carbon-heavy air traffic (Braun-Unkhoff et al., 2017; Trivanovic & Pratsinis, 2024; Zhang et al., 2016). The results of this study imply that the atmospheric



transformation and secondary aerosol formation potential should be taken into consideration when assessing the environmental
impacts of aviation.

**Data availability:** Data are available in the supplementary file or from the authors upon request.

**Author contributions**: **Anni Hartikainen**: Data curation, formal analysis, investigation, visualization, writing – original draft preparation, **Mika Ihalainen:** Investigation, data curation, methodology, **Deeksha Shukla:** data curation, formal analysis, investigation, writing – review and editing, **Marius Rohkamp:** data curation, formal analysis, investigation, project administration, writing – review and editing**, Arya Mukherjee:** investigation, writing – review and editing, **Quanfu He:** investigation, formal analysis, writing – review and editing**, Sandra Piel:** data curation, formal analysis, investigation, 650 visualization writing – review and editing, **Aki Virkkula:** formal analysis, methodology, writing – review and editing**, Delun Li:** investigation, formal analysis, methodology, **Tuukka Kokkola:** data curation, formal analysis, investigation, writing – review and editing, **Seongho Jeong:** data curation, investigation, writing – review and editing**, Hanna Koponen:** investigation, writing – review and editing**, Uwe Etzien**: conceptualization, investigation, **Anusmita Das:** investigation, **Krista Luoma:** methodology, formal analysis**, Lukas Schwalb:** data curation, formal analysis, investigation, visualization**, Thomas Gröger**: 655 methodology, supervision, writing – review and editing, **Alexandre Barth**: investigation, resources**, Martin Sklorz**: investigation, supervision, **Thorsten Streibel**: investigation, supervision, project administration, **Hendryk Czech**: investigation, supervision, writing – review and editing, **Benedikt Gündling**: data curation, formal analysis, investigation, writing – review and editing**, Markus Kalberer**: resources, supervision, **Bert Buchholz**: conceptualization, resources, supervision, funding acquisition, **Andreas Hupfer**: conceptualization, resources, supervision, writing – review and editing, 660 **Thomas Adam**: conceptualization, resources, supervision, funding acquisition**, Thorsten Hohaus**: conceptualization, resources, supervision, writing – review and editing, **Johan Øvrevik**: conceptualization, funding acquisition, writing – review and editing, **Ralf Zimmermann**: conceptualization, funding acquisition, resources, supervision, **Olli Sippula**: conceptualization, funding acquisition, resources, supervision, project administration, writing – review and editing

**Competing interests**: Quanfu He is a member of the editorial board of Atmospheric Chemistry and Physics.

**Acknowledgements**

This work was supported by EU Horizon 2020 project ULTRHAS (agreement 955390) and Research Council of Finland project 'Black and Brown Carbon in the Atmosphere and the Cryosphere (BBrCAC) (grant 341597). H.C. acknowledges funding from Helmholtz International Lab aeroHEALTH (Interlabs-0005).



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
