# Peer review of "Photochemical aging of aviation emissions: transformation of chemical and physical properties of exhaust emissions from a laboratory-scale jet engine combustion chamber"

_EGUsphere, 2024_

## Author Comment (AC1)

We thank the reviewers for their time and valuable comments. We have revised the manuscript accordingly and here provide point-to-point responses to the comments. The original referee comments are given in *blue, italic font*, with our responses given in black. The line numbers in the responses refer to the revised text (without changes visible).

Revised texts are given in the intended texts as in the revised manuscript.

In addition, we have fixed the references to comply with the journal standard, as requested by the editorial support team.

**Anonymous referee #1**

**Summary**

*This manuscript presents an interesting study on the effect of photochemical aging on aviation emissions. Such research is important as aerosol emissions can undergo significant chemical and morphological changes due to aging. This in turn can significantly change the effect of the emissions on human health and the environment. The detailed analysis of the gaseous and particulate emissions with many different methods provides a wealth of information on this topic that makes it worth publishing. However, some improvements could be made to this article in particular by better explaining how the results obtained here compare to the existing literature on this topic and the implications of these results. The conclusion does explain the implications of the results reasonably well, but it would be useful to elaborate on this in the discussion section.*

**Specific comments:**

***Pg. 2 – 3, Introduction:*** *The introduction overall does a good job of summarizing the issue of particulate emissions from aviation however, what seems to be missing is a review of the current literature on photochemical aging of exhaust. A summary of any articles which measure PM surrounding airports would be useful to understand if the results obtained here correlate to real-world observations. In addition, if there are no or few studies on the photochemical aging of aircraft exhaust, similar studies conducted on exhaust from other combustion sources such as vehicles could be useful to put the current work in context with the broader scientific literature.*

The introduction was improved by the addition of following sections on

1) particulate matter near airports (l. 44-49):

> "Air pollution in airports and near-airport areas originates from aircraft exhaust emissions, including pollutants derived from fuel combustion and engine lubricants, and other, mainly on-road traffic related sources (Alzahrani et al., 2024; Ungeheuer et al., 2021). UFP from aviation often dominate the particle number concentration in these areas and based on ambient observations particles from aircraft exhausts can be distinguished from the background urban air specifically by their small size (Hudda et al., 2016; Masiol et al., 2017; Stacey, 2019; Zhang et al., 2020)."

2) secondary aerosol formation from exhaust emissions of vehicular sources in general (l. 60-71):

> "Secondary particle formation may lead to manifold particulate matter emissions for traffic sources utilizing liquid fossil fuels, with the secondary formation potential largely depending on the engine operation mode, fuel choice, and exhaust after-treatment systems (Gentner et al., 2017; Hartikainen et al., 2023; Karjalainen et al., 2022; Kostenidou et al., 2024; Paul et al., 2024; Pieber et al., 2018). Unburned fuel, specifically gaseous aromatic compounds, have been suggested as the main precursors for

secondary organic aerosol (SOA) formation in engine exhausts, but uncertainties remain due to the complex interplays of the varying components of exhaust emissions and ambient air (Gentner et al., 2017; Hartikainen et al., 2023; Karjalainen et al., 2022; Kostenidou et al., 2024; Paul et al., 2024; Pieber et al., 2018). Previous estimates of the secondary particle formation from aircraft engines are scarce but point towards significant SOA formation potential. SOA mass may exceed the amounts of fresh particulate matter by one or two orders of magnitude at low load operation in aircraft exhausts (Kiliç et al., 2018; Miracolo et al., 2011, 2012). There is, however, a lack of detailed chemical-physical characterization of secondary particles produced from aviation exhaust emissions, which would be essential for assessing their environmental health and climate effects."

*Pg. 3, lines 83 – 84: It is not clear what is meant by "An original combustion chamber taken a small jet engine with a net thrust of 180 N was used (Hupfer et al. 2012)." The work cited by Hupfer et al. 2012 is not available online and therefore, the jet engine used should be described in greater detail here. In particular, please explain the ways in which this combustion chamber does or does not represent real world conditions. In the following paragraph, the emissions are compared to commercial aircraft which typically have rated thrusts 1 – 3 orders of magnitude higher than the 180 N produced here. How do you know that the particles produced here are similar enough to that produced by commercial engines?*

The combustion chamber has been described in detail also in a publicly available conference proceedings by Fuchs et al. (2016), which is now cited instead of Hupfer et al. (2012). The comparability of our model system to real (but older) turboshaft engines (namely, an Allison 250 C-20B (Rohkamp et al., 2024) and the ICAO database of in-use engines (International Civil Aviation Organization, 2022)) was validated in the careful preliminary measurements on the test rig. The gaseous emissions, mainly CO and THC, were high but remained in the range observed for older engines, as discussed on l. 361-367. The particle emission indices and particle size were also similar to real engines in idle mode, as discussed in Section 3.2.2.

However, a small-scale laboratory system is always still a model system, and the results may not be directly extrapolated to large real-scale engines. This is now more expressly pointed out in the Conclusions. As highlighted in the conclusion section, the need for studies on the aging of exhaust emissions from large-scale commercial engines is evident. This is especially true now as the engine and fuel technologies are in a state of rapid transition, which will also influence the aerosol emissions (both fresh and aged) of aviation.

The consideration of the scope and limitations related to the use of a model system are now better noted in Conclusions, l. 657-664:

> "This work focused on mimicking the airport operation by replicating the average emissions of an LTO cycle using a laboratory model system, and the reported characteristics simulate on- or near-ground exhaust emissions originating from an airport. The influence of operation conditions on the contents of aging aircraft exhaust emissions should be thoroughly studied in the future. Further, the presented characteristics are for secondary aerosol from kerosene-combustion exhaust emission purely, as the experiments were conducted without lubrication oil. Lubrication oil has been noted to be a prominent precursor for soot or volatile particulate matter formation from full scale turbine engines (Fushimi et al., 2019; Ungeheuer et al., 2022; Yu et al., 2010), and the impacts of the mixing of lubrication oil and kerosene combustion exhausts on the aging products requires further studies."

*Pg. 3, lines 95 – 96: Even for the same type of jet fuel, there can be differences from batch-to-batch that have measurable effects on the resulting aerosol. Here, two types of fuel are reportedly used and an average composition was given. What were the differences between the Jet A1 and JP-8 fuel? Was*

*there any correlation between the fuel used and the resulting particles (i.e. changes in particle number, size, etc.)?*

Only one fuel type (JP-8) was used. This is now rephrased on l. 104-106 to avoid further mix-ups.

> "The fuel in use was kerosene-based tactical air fuel JP-8. This military grade fuel is specified according to Jet A-1 but includes additional additives to enhance anti-icing and -corrosion properties compared to the Jet A-1 most often used in civil aviation."

Fuel of same quality from two barrels had to be used as the first barrel ran empty, and the fuel composition was analyzed only for one of the fuel barrels. The two barrels are, however, suspected to have been of similar quality, and no differences in the exhaust emission could be related to the barrel in use, as now clearly stated on l. 112-113:

> "Fuel of same quality from two barrels was used during the campaign; no differences in the exhaust emissions could be related to the barrel in use."

***Pg. 4, lines 105 – 108:*** *The text in Figure 1, particularly the numbers on the GC x GC-MS chromatographs, is too small to read. Please increase the size of the text so that it is legible. If text such as the numbers on the chromatographs is not important due to the qualitative nature of the data, then remove the text rather than leaving too-small-to-read characters on the graph.*

We have increased the font size and removed the too-small-to-read numbers from the chromatographs, which indeed aim to provide qualitative information of the data.

***Pg. 4, lines 110 – 120:*** *How long are the different sections of the sampling lines? What corrections, if any, have been made to account of diffusion losses in the sampling lines?*

The exhaust line length to the heated dilutor system was eight meters (⌀ 80 mm), after which there was a sampling line of eight meters (⌀ 40 mm) to the flow reactor/bypass line, before dividing the sample to the aerosol instrumentation. The experimental setup was designed with the aim of keeping the particle losses to the minimum within the constraints of the setup, and no corrections were made to account for losses by diffusion. Due to the high flow rate the sample flow in the sampling lines was turbulent, but simultaneously the particle losses were minimized by keeping the residence time as short as possible. Using general diffusion loss estimations for turbulent flow (Baron & Willeke, 2001), the losses of the smallest particles measured by the SMPS (7 nm) were estimated to remain below 10 % (below 1 % for 100 nm particles) in the ⌀ 40 mm sampling line after the dilutors, where we expect the majority of the largely organic particles to have formed via condensation and nucleation of the gaseous species. Particle losses in the PEAR are minimized by the laminar flow and design of the flow reactor (Ihalainen et al., 2019).

The setup is now clarified in text (l. 121-129):

> "The exhaust tube (⌀ 80 mm) length from the jet engine burner outlet to the sampling point was eight meters. Exhaust was sampled to the aerosol instrumentation through a heated precyclone (400 ℃) and a heated line (350 ℃) and diluted by combination of a porous tube diluter (PTD) and ejector diluter (ED) (DAS, Venacontra, Finland) to minimize particle losses during sample cooling. The sampling line length downstream the dilutor system was eight meters (⌀ 40 mm) to the PEAR/bypass line, before the sample was divided to the aerosol instrumentation. Diffusion losses were estimated using generally available diffusion loss estimations for turbulent flow (Baron & Willeke, 2001) and kept to the minimum by constraining residence time of the turbulent sample flow by the high flow rate. The estimated diffusion losses remained below 10 % for the smallest particles measured (7 nm) and below 1 % for 100 nm particles. No corrections to measurement data were made to account for the losses in the experimental system."

***Pg. 6, line 149:*** *Could you elaborate on how "...careful conditioning of the sample..." was achieved? What exactly was done to condition the sample?*

The sample dilution was set to a sufficient range and the temperature of the heated line before the dilution was adjusted to maintain the sample temperature before dilution with clean air at room temperature. Additionally, the amounts of water and ozone added to the PEAR along the sample were set to obtain representative aging conditions at the selected UV intensity and flow rate. This is now clarified on l. 168-170:

> "Namely, sufficient dilution was selected to decrease external OH reactivity in sample stream and water vapor concentration, O3 input, and UV intensity were set to establish conditions for tropospherically relevant photochemical processing."

***Pg. 12, lines 342 – 343:*** *What is approximately the range of pressures that can be expected in newer vs. older engines? How does this compare to the pressure in your set-up?*

Pressures in our system were lower than in real jet engines, and are reported in Supplementary Figure S1. This is now also clearly mentioned in Methods (l.102-103).

> "The pressure levels in the combustor system were generally lower than for real jet engines (Fig. S1). More information on the test rig operation conditions is available in Fig. S1."

Typical maximum overall pressure ratios (the ratio of the total pressure at the compressor outlet to the total pressure at the compressor inlet) in older engines are approx.10 for old, approx. 50 for newer engines (see, e.g., ICAO databank available at https://www.easa.europa.eu/en/domains/environment/icao-aircraft-engine-emissions-databank). However, on ground-idle the OPRs in both older and newer engines are significantly lower (~2 – 15, depending on technology; not routinely publicly reported) than in cruise or take-off, as the compressor doesn't operate at maximum speed and the mass flow is reduced. The compression in our engine was lower than even in older engine technologies; OPR can't be conceptually defined for this model system due to its differences to a real jet turbine engine, but the pressures used during the experiments represent OPRs above 1.

Sentence on l. 362-365 was rephrased to

> "Older engines also do not have high overall pressure ratios (OPR), whereas in newer engines the gaseous emissions are typically reduced by the higher compression, temperatures, and higher combustion efficiency. The OPR of newer engines is also lower in idling and taxiing conditions..."

***Pg. 12, section 3.1.2:*** *Can you comment on the implications of the changes in VOC EI after photochemical processing?*

Concentrations of VOCs susceptible for photochemical reactions are bound to change in the atmosphere (and in our laboratory experiments) by photolysis and reactions with OH, as they are oxidized (and partly fragmented) into other gaseous and condensable species. Some of the VOCs ultimately form particulate matter with the consequent change in volatility, as shown in our study.

Several of the observed VOCs are themselves harmful to health, and changes in their concentrations may alter the air quality implications of the exhaust emissions. For example, the formation of gas phase carbonyls, which are known to induce toxic effects, and decay of gaseous PAHs (and subsequent formation of particulate PAHs) may alter the health implications of the exhaust emissions. This has been stated in Conclusions, l. 655-657:

"The observed secondary formation of gas-phase carbonyls, transformation in the polycyclic aromatic content, and the increased oxidation state of the organic particles can also alter the toxicity of the exhaust emissions."

PAHs and carbonyls have been noted as toxic air pollutants, and the potential health implications of changes in their EIs are now also noted in the Result section 3.1.2., l. 410-412:

"Changes in the emissions of these gaseous compounds may alter the harmful health impacts in ambient air, as PAHs and several carbonylic compounds have been identified as toxic air pollutants with, e.g., carcinogenic or mutagenic effects (Clergé et al., 2019; Eder et al., 1990; IARC, 2025; Zhang et al., 2019)."

***Pg. 14, lines 395 – 397:** Why do you think the particles produced here are more spherical than those observed in studies from real aircraft engines? If the particles are substantially different from what is observed in real engines, what differences might we expect to see in the aging of real aircraft exhaust?*

The combustion conditions in our combustor led to exhaust emissions with minor amounts of soot compared to organic emissions, as typical for low-power operation of aircraft engines. The condensable organic species were also included in our experiments, as our sampling system dilutes and cools the sampled exhaust aerosol with the aim of realistically mimicking processes that occur when the exhaust gases are emitted into ambient air. This led to 1) formation of round nucleated organic particles and 2) addition of condensing coating on fractal agglomerates, which is known to lead to agglomerate compaction (Leskinen et al., 2023; Sipkens & Corbin, 2024). This compaction of the few soot agglomerates present in the exhaust was observed also for our setup: in the pre-campaign experiments dedicated to finding the correct engine setpoints, EM grids were collected from 0.5 meters behind the combustion chamber, and the scanning electron micrographs (Fig. R1) show the initial particles to have been slightly less round than in the diluted and cooled exhaust, albeit non-fractal compared to, e.g., Durdina et al. (2014) or Saffaripour et al. (2020). The shape is, however, very similar to that of particles from an older turboshaft engine (Allison 250 C-20B) in idle operation mode (Rohkamp et al., 2024).

The reasons for the observed sphericality are now explained in more detail on l. 421-424:

"This discrepancy is caused by the scarcity of soot agglomerates relative to the condensable organic species, which led to 1) formation of round nucleated organic particles and 2) addition of condensing coating on the particles, which is known to lead to agglomerate compaction (Leskinen et al., 2023; Sipkens & Corbin, 2024)."

[Figure]

**Figure R1.** *Scanning electron micrographs of exhaust particles in the fresh emission, sampled 0.5 m after the combustor rig in pre-campaign experiments.*

**Pg. 14, lines 402 – 403:** *What was average aerodynamic size of the fresh particles for comparison? Based off the rest of your analysis, do you know how the particles are growing? In other words, which compounds caused the growth and what was the growth mechanism?*

Aerodynamic size of the fresh particles was not directly measured by any instrument. It can however be estimated for the approximately round particles based on the known mobility sizes ($d_m$) and particle effective density as $d_{Ae} = d_m / \sqrt{\frac{\rho_0}{\rho_p}}$. Unfortunately, the particle density couldn't be defined for the smallest fresh particles due to their scarcity, but assumably fell between 1 and 1.5 g/cm³. This would mean that, for example, the aerodynamic modes of the fresh particle size distributions (13 and 25 nm in mobility diameter) would either be identical to the mobility diameter or up to 16 or 30 nm, respectively.

The particles grew by condensational growth caused by the oxidation and consequent condensation of the initially gaseous precursor species. All in all, several compounds cause the formation of condensing material, as discussed in Section 3.4. The volatile organic compounds quantified by the PTR-TOF were estimated to account for only approximately one quarter of the observed secondary mass, meaning that we couldn't resolve most of the culprits for secondary particle formation. Based on the rest of the analysis we hypothesize that the role of the cycloalkanes could be important, but can't be quantified based on this work. The particle concentrations in the PEAR were very low, so growth by coagulation in the PEAR was negligible (Fig. R2). This has been stated on l. 456-457.

[Figure]

***Figure R2.*** *Particle number size distributions at the PEAR input and the estimated shifts in particle size distribution due to coagulation only in the PEAR. Coagulation was modelled using the Fuchs form of Brownian coagulation (Seinfeld & Pandis, 2016) and the diluted size distributions measured by SMPS upstream the PEAR (shown in black).*

**Pg. 15, lines 409 – 411:** *This sentence is hard to understand because of the complicated grammatical structure. Consider rephrasing.*

This sentence was rephrased (l. 435-438):

> "The particle size distributions (PSDs) of the fresh samples were bimodal, with modes at 13 nm and 25 nm (Fig. 5). The fresh PSDs were influenced by the sampling and dilution settings, which affected the coagulation and gas-particle partitioning of the cooling exhaust emission prior to the SMPS measurements."

**Pg. 16, lines 435 – 440:** *You state that the LDSA was 370 cm²/kg$_{fuel}$⁻¹ for fresh emissions and 8900 cm²/kg$_{fuel}$⁻¹ for aged emissions. However, you also state that "…for the size range of the fresh exhaust*

*particles, any particle growth would decrease their deposition efficiency..." If the particles grew in size with aging as shown in Fig. 5, why did the LDSA increase? Is the increased LDSA due to the increased number of particles? Perhaps this is a misunderstanding, if so, please consider rephrasing this text to make the meaning clearer.*

Deposition efficiency describes the probability that a particular particle is deposited on the human respiratory track. For the ultrafine size region of the particles observed in this study, this probability decreases with increasing size. The LDSA indeed grew because of the massive increase in the total amount of particles, but less than the relative increase in mass. This paragraph (l. 462-469) aims to explain why the relative increase in LDSA is so much lower than the increase in particle mass, and has been rephrased to:

> "The alveolar LDSA increased with the vast secondary aerosol surface formation with an enhancement ratio of 24, which is distinctly below the enhancement in total particulate mass. This difference in the relative increase is due to the size dependency of lung deposition efficiency: in the alveolar region, the deposition efficiency of individual particles peaks at 50 % at 20 nm and is below 20 % for particles above 50 nm in mobility diameter (Hinds & Zhu, 2022). This means that for the size range of the fresh exhaust particles, any particle growth would decrease their individual deposition efficiency in the lungs and head airways."

***Pg. 23, lines 600 – 602:** If the new particle mass formation cannot be explained by the oxidation of volatile precursors, what other potential mechanisms exist to explain the new particle mass formation observed?*

The new particle mass was formed by the oxidation of the organic volatile/semi/intermediately volatile species initially in the gas phase. The only other potential mechanism for the observed mass increase might arise from growth of the particles that were initially too small to measure with our setup (<7 nm) to a measurable size range; however, considering the measured size distributions, addition of mass by this mechanism was negligible.

Unfortunately, it is evident that our gas phase analyses did not cover all the potential precursors. For example, we didn't include the potentially important cycloalkanes in the offline analyses and overall missed the gas-phase low-volatility species with our setup. In Section 3.4., we hypothesize that these species may have played an important role in the secondary particle formation.

***Pg. 24, lines 626 – 633:** While PM can radiative forcing effects, it is estimated to be a very small proportion of aviation's net-radiative forcing (https://doi.org/10.1016/j.atmosenv.2020.117834). Given this, do the changes in RF observed here represent a significant enough change to influence local warming or cooling?*

These results cannot by themselves be used for estimating the influence of aviation emissions on local cooling/warming, as they describe the relative potential for individual particles from one source. The local radiative forcing is influenced by emissions from many sources, with their relative significance also depending on the prevalence of each source. However, our results give insights on how secondary aerosol formation may impact the radiative forcing properties of these specific exhaust emissions, and show that the aging of the emissions must be considered when estimating the environmental implications of aviation.

***Pg. 24, lines 634 – 636:** How do the results observed here compare to measurements of ambient air around airports?*

Ambient air measurements near airports routinely report that aviation emission dominate the total particle concentration and produce the smallest particles observed in urban air in- or near airports.

The notion of this in the Introduction has now been revised (l. 44-49), as shown in the reply to the first comment.

In contrast to direct plume measurements, which often include only the non-volatile particle fraction, ambient measurements typically include also volatile particles. Our results show that majority of this volatile fraction may actually originate from in-air processing of the exhaust rather than directly from the source. We discuss this in the Conclusion section (l. 650-652):

> "This indicates that the aviation-derived ambient ultrafine particles can originate also from photochemical processing of originally gaseous organic compounds, and that secondary organic aerosol from aircraft exhaust emissions can be a notable source of organic particulate matter, including UFPs, to urban air."

The distinguishing of the chemical composition of the smallest UFPs is hindered by their small size: although they may dominate in particle number, the volume fraction is easily superseded by other sources. The distinguishing of chemical composition of aviation-originating aerosols from, e.g., filter samples collected from ambient air is consequently difficult. Simultaneously, the particles are typically too small for common size-selective online chemical measurements (e.g., aerosol mass spectrometry). This also impedes the direct comparison of our results with ambient observations.

References:

Alzahrani, S., Kılıç, D., Flynn, M., Williams, P. I., & Allan, J. (2024). International airport emissions and their impact on local air quality: chemical speciation of ambient aerosols at Madrid–Barajas Airport during the AVIATOR campaign. Atmospheric Chemistry and Physics, 24(16), 9045–9058. https://doi.org/10.5194/acp-24-9045-2024

Baron, P. A., and Willeke K. (Eds.): Aerosol measurement: principles, techniques, and applications (Second edition). John Wiley & Sons, United States of America. ISBN 0471356360, 2001.

Clergé, A., Le Goff, J., Lopez, C., Ledauphin, J., & Delépée, R. (2019). Oxy-PAHs: occurrence in the environment and potential genotoxic/mutagenic risk assessment for human health. Critical Reviews in Toxicology, 49(4), 302–328. https://doi.org/10.1080/10408444.2019.1605333S

Durdina, L., Brem, B. T., Abegglen, M., Lobo, P., Rindlisbacher, T., Thomson, K. A., Smallwood, G. J., Hagen, D. E., Sierau, B., & Wang, J. (2014). Determination of PM mass emissions from an aircraft turbine engine using particle effective density. Atmospheric Environment, 99, 500–507. https://doi.org/10.1016/j.atmosenv.2014.10.018

Gentner, D. R., Jathar, S. H., Gordon, T. D., Bahreini, R., Day, D. A., El Haddad, I., Hayes, P. L., Pieber, S. M., Platt, S. M., de Gouw, J., Goldstein, A. H., Harley, R. A., Jimenez, J. L., Prévôt, A. S. H., & Robinson, A. L. (2017). Review of Urban Secondary Organic Aerosol Formation from Gasoline and Diesel Motor Vehicle Emissions. Environmental Science & Technology, 51(3), 1074–1093. https://doi.org/10.1021/acs.est.6b04509

Eder, E., Hoffman, C., Bastian, H., Deininger, C., & Scheckenbach, S. (1990). Molecular mechanisms of DNA damage initiated by alpha, beta-unsaturated carbonyl compounds as criteria for genotoxicity and mutagenicity. Environmental Health Perspectives, 88, 99–106. https://doi.org/10.1289/ehp.908899

Fuchs, F., Meidinger, V., Neuburger, N., Reiter, T., Zündel, M., & Hupfer, A. (April 2016). Challenges in designing very small jet engines - fuel distribution and atomization. International Symposium on Transport Phenomena and Dynamics of Rotating Machinery. Honolulu, United States. https://hal.science/hal-01891309v1

Hartikainen, A. H., Ihalainen, M., Yli-Pirilä, P., Hao, L., Kortelainen, M., Pieber, S. M., & Sippula, O. (2023). Photochemical transformation and secondary aerosol formation potential of Euro6 gasoline and diesel

passenger car exhaust emissions. Journal of Aerosol Science, 171, 106159. https://doi.org/10.1016/j.jaerosci.2023.106159

Hudda, N., Simon, M. C., Zamore, W., Brugge, D., & Durant, J. L. (2016). Aviation Emissions Impact Ambient Ultrafine Particle Concentrations in the Greater Boston Area. Environmental Science & Technology, 50(16), 8514–8521. https://doi.org/10.1021/acs.est.6b01815

IARC. (2025). Agents classified by the IARC Monographs, Volumes 1–137. https://monographs.iarc.who.int/list-of-classifications.

Ihalainen M., et al. (2019). A novel high-volume Photochemical Emission Aging flow tube Reactor (PEAR). Aerosol Science and Technology, 53(3), 276–294. https://doi.org/10.1080/02786826.2018.1559918

International Civil Aviation Organization. (2022). ICAO Engine Emissions Databank. Available in electronic format from the EASA website: https://www.easa.europa.eu/en/domains/environment/icao-aircraft-engine-emissions-databank

Karjalainen, P., Teinilä, K., Kuittinen, N., Aakko-Saksa, P., Bloss, M., Vesala, H., Pettinen, R., Saarikoski, S., Jalkanen, J.-P., & Timonen, H. (2022). Real-world particle emissions and secondary aerosol formation from a diesel oxidation catalyst and scrubber equipped ship operating with two fuels in a SECA area. Environmental Pollution, 292, 118278. https://doi.org/10.1016/j.envpol.2021.118278

Kostenidou, E., Marques, B., Temime-Roussel, B., Liu, Y., Vansevenant, B., Sartelet, K., & D'Anna, B. (2024). Secondary organic aerosol formed by Euro 5 gasoline vehicle emissions: chemical composition and gas-to-particle phase partitioning. Atmospheric Chemistry and Physics, 24(4), 2705–2729. https://doi.org/10.5194/acp-24-2705-2024

Leskinen, J., Hartikainen, A., Väätäinen, S., Ihalainen, M., Virkkula, A., Mesceriakovas, A., Tiitta, P., Miettinen, M., Lamberg, H., Czech, H., Yli-Pirilä, P., Tissari, J., Jakobi, G., Zimmermann, R., & Sippula, O. (2023). Photochemical Aging Induces Changes in the Effective Densities, Morphologies, and Optical Properties of Combustion Aerosol Particles. Environmental Science & Technology, 57(13), 5137–5148. https://doi.org/10.1021/acs.est.2c04151

Masiol, M., Harrison, R. M., Vu, T. V., & Beddows, D. C. S. (2017). Sources of sub-micrometre particles near a major international airport. Atmospheric Chemistry and Physics, 17(20), 12379–12403. https://doi.org/10.5194/acp-17-12379-2017

Paul, A., Fang, Z., Martens, P., Mukherjee, A., Jakobi, G., Ihalainen, M., Kortelainen, M., Somero, M., Yli-Pirilä, P., Hohaus, T., Czech, H., Kalberer, M., Sippula, O., Rudich, Y., Zimmermann, R., & Kiendler-Scharr, A. (2024). Formation of secondary aerosol from emissions of a Euro 6d-compliant gasoline vehicle with a particle filter. Environmental Science: Atmospheres, 4(7), 802–812. https://doi.org/10.1039/D3EA00165B

Pieber, S. M., Kumar, N. K., Klein, F., Comte, P., Bhattu, D., Dommen, J., Bruns, E. A., Kılıç, D., El Haddad, I., Keller, A., Czerwinski, J., Heeb, N., Baltensperger, U., Slowik, J. G., & Prévôt, A. S. H. (2018). Gas-phase composition and secondary organic aerosol formation from standard and particle filter-retrofitted gasoline direct injection vehicles investigated in a batch and flow reactor. Atmospheric Chemistry and Physics, 18(13), 9929–9954. https://doi.org/10.5194/acp-18-9929-2018

Rohkamp, M., Rabl, A., Bendl, J., Neukirchen, C., Saraji-Bozorgzad, M. R., Helcig, C., Hupfer, A., & Adam, T. (2024). Gaseous and particulate matter (PM) emissions from a turboshaft-engine using different blends of sustainable aviation fuel (SAF). Aerosol Science and Technology, 1–16. https://doi.org/10.1080/02786826.2024.2417977

Saffaripour, M., Thomson, K. A., Smallwood, G. J., & Lobo, P. (2020). A review on the morphological properties of non-volatile particulate matter emissions from aircraft turbine engines. Journal of Aerosol Science, 139, 105467. https://doi.org/10.1016/j.jaerosci.2019.105467

Seinfeld, John H., and Spyros N. Pandis. Atmospheric chemistry and physics: from air pollution to climate change. John Wiley & Sons, 2016.

Sipkens, T. A., & Corbin, J. C. (2024). Effective density and packing of compacted soot aggregates. Carbon, 226, 119197. https://doi.org/10.1016/j.carbon.2024.119197

Stacey, B. (2019). Measurement of ultrafine particles at airports: A review. Atmospheric Environment, 198, 463–477. https://doi.org/10.1016/j.atmosenv.2018.10.041

Ungeheuer, F., van Pinxteren, D., & Vogel, A. L. (2021). Identification and source attribution of organic compounds in ultrafine particles near Frankfurt International Airport. Atmospheric Chemistry and Physics, 21(5), 3763–3775. https://doi.org/10.5194/acp-21-3763-2021

Zhang, X., Karl, M., Zhang, L., & Wang, J. (2020). Influence of Aviation Emission on the Particle Number Concentration near Zurich Airport. Environmental Science & Technology, 54(22), 14161–14171. https://doi.org/10.1021/acs.est.0c02249

Zhang, Y., Xue, L., Dong, C., Wang, T., Mellouki, A., Zhang, Q., & Wang, W. (2019). Gaseous carbonyls in China's atmosphere: Tempo-spatial distributions, sources, photochemical formation, and impact on air quality. Atmospheric Environment, 214, 116863. https://doi.org/10.1016/j.atmosenv.2019.116863

*Hartikainen et al. presented a comprehensive study on the photochemical aging of emissions from a laboratory-scale jet engine with a focus on the transformation of organic species and absorption properties of secondary organic aerosols formed. Useful dataset on aging of aviation exhausts are presented in this manuscript, which surely expand our understanding on an important source of atmospheric pollutants. On the other hand, this reviewer believe that the authors should further clarify their experimental settings and relevant uncertainties to increase the robustness of the conclusions.*

1. *It is well known that flow tube experiments are characterized with branching ratios of reaction channels different from those in the ambient, which subsequently impact the yields of secondary organic aerosols, the chemical identities, and potentially the absorption properties of aerosol particles. Although such a deviation from the ambient is unavoidable, the authors are suggested to give a discussion, e.g., initial organic/NOx ratios, branching ratios, and any enhanced products that may contribute to absorption.*

The initial ratio of the total gaseous hydrocarbons (THC, measured by FID) to $NO_x$ was 70 ppmC ppm$^{-1}$ in the fresh exhaust emission, which makes the SOA formation conditions $NO_x$-limited. This has been stated on l. 539. Mass absorption efficiency of SOA has been noted to generally depend on $NO_x$ availability, so that lower MAEs have been measured in $NO_x$ limited conditions where no formation of, e.g., nitroaromatics (effective light absorbers) occurs (He et al., 2020, Liu et al., 2017; Nakayama et al., 2013).

In previous work comparing the effects of aging conditions in the PEAR versus an environmental chamber (Czech et al., 2024), the optical properties of SOA (namely, MAE or AAE) of toluene-derived SOA in $NO_x$-limited conditions were not altered by aging conditions; this doesn't mean that it can't impact it for other precursors. The optical properties of SOA formed in our study are likely to have been affected by the mixing of various precursors, which have non-linear effects on the optical properties (Cui et al., 2024), complicating the comparison with previous studies where most often single precursors have been in use.

These potential impacts are now noted on l. 602-608:

> "The mixing of SOA precursors is likely to have non-linear effects on optical properties (Cui et al., 2024), highlighting the need for studies of real-life mixtures of precursor classes.
>
> The MAE of SOA also depends on the aging conditions, such as RH and NOx availability (He et al., 2020, Liu et al., 2017; Nakayama et al., 2013). It is likely that higher MAEs would have been observed in case of lower VOC/NOx ratio. The relatively photolysis-rich aging conditions may also alter the reaction pathways and, consequently, properties of the forming SOA. Previously, no OFR-condition dependent differences in optical properties were observed for toluene-derived SOA in no-NOx conditions (Czech et al., 2024), but differences for SOA from other precursors can't be ruled out."

2. *I assume that the black line was heated to 350 degree whereas the green and yellow dashed lines are not. So "the fresh" later in the manuscript stands for emissions after cooling and dilution? Maybe, also through the PEAR to take into account any condensation/loss in the flow tube?*

This is correct, only the line from the pre-cut impactor (400 ℃) to the PRD+ED dilution setup was heated to 350 ℃. The lines following the dilutors were at room temperature. The gas phase measurements with FTIR and FID were done from the raw exhaust gas prior to dilution, as shown in Fig. 2, which is now revised to better show the heated section of the lines.

For all the particle phase instrumentation, 'fresh' indeed refers to emissions after cooling and dilution. The diluted fresh emissions were sampled through a bypass line (i.e., not through PEAR). We have now revised Figure 2 to better represent the experimental setup and sampling of fresh/aged exhaust:

[Figure]

Additionally, the explanation of the sampling setup in text was improved (l. 121-129):

> "The exhaust tube (⌀ 80 mm) length from the jet engine burner outlet to the sampling point was eight meters. Exhaust was sampled to the aerosol instrumentation through a heated precyclone (400 °C) and a heated line (350 °C) and diluted by combination of a porous tube diluter (PTD) and ejector diluter (ED) (DAS, Venacontra, Finland) to minimize particle losses during sample cooling. The sampling line length downstream the dilutor system was eight meters (⌀ 40 mm) to the PEAR/bypass line, before the sample was divided to the aerosol instrumentation. Diffusion losses were estimated using generally available diffusion loss estimations for turbulent flow (Baron & Willeke, 2001) and kept to the minimum by constraining residence time of the turbulent sample flow by the high flow rate. The estimated diffusion losses remained below 10 % for the smallest particles measured (7 nm) and below 1 % for 100 nm particles. No corrections to measurement data were made to account for the losses in the experimental system."

3. Also, as the authors stated (Line 216), there was a potential issue to install QFFs in front of the adsorber tubes. I would like to see a more quantitative analysis on the accuracy of quantification of organics using this setup, and consequently, the results derived from these measurements.

QFFs were placed before the adsorption tubes as pre-filters to avoid overloading the adsorber tubes, and these shouldn't impact the analysis of the measured VOCs. In practice, a stainless-steel filter holder assembly was positioned between an empty glass tube upstream and the adsorber tube downstream. This filter holder held 13 mm QFFs (precipitation area diameter of 10 mm) punched from separately baked 47 mm QFF filters. This setup has been validated to provide reliable results for combustion aerosols specifically, as described in detail in Mason et al. (2020). This information has now been clarified in the Methods (l. 145-148) and in the Supplementary section S1.2.

The section referred to by the reviewer (now line 236) discusses the impact of volatile species on the filter analyses. The QFFs used for the GC×GC-analyses were collected from different line than the adsorber tubes; this is now clarified on l. 146 to avoid possible future confusions. Because of the volatility of the material sampled on the filters, we avoid giving quantitative data from the GC×GC HR-ToF-MS analyzes. However, we consider the presented semi-quantitative data to provide important information on the exhaust particles composition.

Revised text on l. 145-148 now reads:

> "Gas-phase organics of different volatilities were collected on adsorber tubes with three sublayers of Graphitized Carbon Black (GCB) sorbents (Table S1; Supplementary section 1.2) from a parallel sampling line. Adsorber tubes were conditioned under a protective nitrogen atmosphere at 350 ℃ for 1 h 30 min. Baked QFFs punches (13 mm) were installed in front of the adsorber tubes to remove particle fractions from the sample flow (Mason et al., 2020)."

4. (Line 260), did you see any negative value from the subtraction?

No, the OC measured from the filters following extraction was consistently lower than the original OC.

5. The authors made a number of assumptions during the calculation, e.g., a multiple scattering correction factor of 3.4 (Line 273), identical core volume fractions for all particle sizes (Line 305). A further justification is suggested.

The multiple correction factor of 3.4 was used to better compare the MACs defined directly from QFF filters to 1) aethalometer-based estimates and 2) MACs from the water extracts. We are aware of the uncertainty this brings to the absolute values and discuss this in the Method section (l. 292-299). Note, however, that the MACs from the direct filter measurements were not used for the RFE calculation. In the revision, a further note regarding the effects of using one selected multiple scattering correction for filter-based absorption was added to Result Section 3.5. (l. 575-576)

> "It should be noted that the use of a single correction factor for multiple scattering introduces an uncertainty to these values, because the filter effects by the remaining particles may have differed at different extraction stages."

The core volume fraction typically varies depending on particle size. In addition, the condensing material would in reality likely form particles of different sizes and with different core fractions than in our experiments, where all the sampled aerosol consisted of the exhaust emissions of our combustor only. Unfortunately, we were unable to resolve the exact core fractions in different situations so we had to assume identical core volume fractions across all size. To cover a range of possible situations, we decided to consider a relatively large range of core fractions (0.01% to 10%) in the Mie model (results shown in e.g., Fig. 8 for 0.02% to 10 %). This analysis helped us showcase how the observed particle formation of the weakly absorbing organic carbon would impact the overall RFEs.

Further explanation for assuming identical core fractions was added on l. 324-328:

> "In reality, the core volume fractions typically vary depending on particle size. However, because the exact core fractions in different situations couldn't be resolved, identical core fractions were assumed for all particle sizes. This is unrealistic, but since shell thickness was not measured, we considered it a reasonable assumption for the modeling, where several volume fractions were considered, and a mixture of shell thicknesses would fall between the modeled cases."

6. (Line 340-345), what does "pressure" refer to?

Overall pressure ratio, i.e., the ratio of the total pressure at the compressor outlet to the total pressure at the compressor inlet. This has now been rephrased (l. 362-365):

> "Older engines also do not have high overall pressure ratios (OPR), whereas in newer engines the gaseous emissions are typically reduced by the higher compression, temperatures, and higher combustion efficiency. The OPR of newer engines is also lower in idling and taxiing conditions..."

7. My calculation indicates that 1 eqv. d = $1.5 \times 10^6$ cm$^{-3}$ x 24 hr by the authors, which is suggested to be stated in the manuscript. Many people would prefer 12 hr in this definition. This

also leads to my concern whether it makes sense to have such a long OH exposure if what we really want to know is the physicochemical properties of aged emission.

The referee is correct in their calculation; the use of 24h as one day is now clarified on l. 160. OFR studies typically reach relatively high OH exposures as they aim to obtain the 'secondary aerosol formation potential' of the sample. Particulate emissions may have lifetimes of weeks in the atmosphere, which is why we also look into a range of exposures in this range. As we saw majority of the SOA potential to have realized within the first 2 eqv.d., this was a reasonable setpoint for the experiments with such long OH exposures.

References:

Cui, Y., Chen, K., Zhang, H., Lin, Y.-H., & Bahreini, R. (2024). Chemical Composition and Optical Properties of Secondary Organic Aerosol from Photooxidation of Volatile Organic Compound Mixtures. ACS ES&T Air, 1(4), 247–258. https://doi.org/10.1021/acsestair.3c00041

Czech, H., Yli-Pirilä, P., Tiitta, P., Ihalainen, M., Hartikainen, A., Schneider, E., Martens, P., Paul, A., Hohaus, T., Rüger, C. P., Jokiniemi, J., Zimmermann, R., & Sippula, O. (2024). The effect of aging conditions at equal OH exposure in an oxidation flow reactor on the composition of toluene-derived secondary organic aerosols. Environmental Science: Atmospheres, 4(7), 718–731. https://doi.org/10.1039/D4EA00027G

He, Q., Li, C., Siemens, K., Morales, A. C., Hettiyadura, A. P. S., Laskin, A., & Rudich, Y. (2022). Optical Properties of Secondary Organic Aerosol Produced by Photooxidation of Naphthalene under NOx Condition. Environmental Science & Technology, 56(8), 4816–4827. https://doi.org/10.1021/acs.est.1c07328

Liu, J., Lin, P., Laskin, A., Laskin, J., Kathmann, S. M., Wise, M., Caylor, R., Imholt, F., Selimovic, V., & Shilling, J. E. (2016). Optical properties and aging of light-absorbing secondary organic aerosol. Atmospheric Chemistry and Physics, 16(19), 12815–12827. https://doi.org/10.5194/acp-16-12815-2016

Nakayama, T., Sato, K., Matsumi, Y., Imamura, T., Yamazaki, A., & Uchiyama, A. (2013). Wavelength and NOx dependent complex refractive index of SOAs generated from the photooxidation of toluene. Atmospheric Chemistry and Physics, 13(2), 531–545. https://doi.org/10.5194/acp-13-531-2013

Mason, Y. C., et al. (2020). Comparative sampling of gas phase volatile and semi-volatile organic fuel emissions from a combustion aerosol standard system. Environmental Technology & Innovation, 19, 100945. https://doi.org/10.1016/j.eti.2020.100945

---

## Author Response (AR2)

We thank the reviewer for the questions and suggestions for improvement. We have revised the manuscript accordingly as shown in these point-by-point responses.

The original referee comments are given in *blue, italic font*. Our responses are given in black. The numbers in the responses refer to the revised text (without changes visible).

Manuscript quotations are given in the intended sections as in the revised text.

*Line 79: It may be more precise to acknowledge certain constraints, as the light-absorbing properties vary among different types of organic aerosols and are not universally significant.*

Both the light absorbing and light scattering properties of organic aerosols may indeed vary depending on the aerosol composition. This is now noted in the Introduction (l. 77-79):

"Optical properties can vary greatly among different types of organic aerosols, and ultimately the direct radiative forcing caused by aviation particles is impacted by absorption and scattering caused by both the primary BC and the potentially BrC-containing organic aerosol formed during the exhaust cooling or in the atmosphere."

*Line 97: Is the combustion chamber used in the study representative enough to make the results broadly applicable to real-world aviation emissions?*

With our setup, we aimed to generate aircraft engine -like exhaust emissions using a laboratory scale combustion chamber in order to estimate the tendencies in photoaging. These results are not broadly applicable to real-world aviation emissions, where the emissions also differ due to the technical design and operating conditions between different engine technologies and generations (see, e.g., the ICAO database at https://www.easa.europa.eu/en/domains/environment/icao-aircraft-engine-emissions-databank).

We have improved the Conclusions of our work when highlighting the need for further studies with different engine models and operation conditions (l. 662-664):

"This work focused on mimicking the airport operation by replicating the average emissions of an LTO cycle using a laboratory model system, to simulate on- or near-ground exhaust emissions originating from an airport. The influence of engine model and operation conditions on the contents of aging aircraft exhaust emissions should be thoroughly studied in the future."

*Line 176: "However, but the photolysis was slightly higher compared to tropospheric conditions, which may have influenced the formation of the oxidation products." Given this, could the authors provide some indication of how this might have influenced their results? Are the findings still considered representative of actual atmospheric aging?*

Photolysis and OH-radical initiated oxidation reactions of organic compounds in the atmosphere are known to lead to different peroxy radical ($RO_2$) reaction pathways and reaction products (see, e.g., Peng & Jimenez, 2020). In general, the fates of $RO_2$ radicals (which are very relevant in atmospheric oxidation processes) are influenced by the extent of OH radical initiated oxidation reactions compared to other reaction pathways. These reaction pathways can vary strongly between different compound groups. The kerosene combustion exhaust used in this study is a complex and partly unresolved mixture of organic compounds, which makes it impossible to evaluate the role of potential

variance in the radical chemistry (such as the over-emphasized photolysis) in the oxidation flow reactor.

Conditioning of a comprehensive flow reactor study requires compromises when adjusting photochemical aging conditions and suitable concentrations for all instruments and concurrent cell exposures. We utilized the framework described by Peng & Jimenez (2017) to guarantee sufficiently representative flow reactor conditions. This is mentioned on l. 166-167:

> "The ratio of photolysis to OH exposure remained between 0.6 and 2 × 10$^6$ cm s$^{-1}$ (Table S2), indicating that in total, the flow reactor conditions were in the 'risky' regime as defined by Peng and Jimenez (2017)."

*Line 225: What collection efficiency was adopted for AMS measurements? Also, what is the degree of agreement (or correlation) between the AMS + aethalometer and SMPS mass measurements?*

AMS CE of 1 was used. This was validated by measuring standard ammonium nitrate and ammonium sulfate particles. This is now mentioned at l. 223:

> "Ionization efficiencies of the HR-ToF-AMS were obtained by measuring 350 nm ammonium nitrate and ammonium sulfate particles. Collection efficiency of 1 was applied."

Mass emission indices derived from the AMS and SMPS measurements are shown side by side in Figure 6. The mass-EIs agreed well for the DR50 experiments, where the particles were in a size range measurable by the AMS. For the DR200 experiments, there was a slight underestimation by the AMS, caused largely by the particle size being below the instrument detection range.

Aethalometer doesn't measure mass but light attenuation. The 'equivalent' masses are often calculated using a known/set mass absorption efficiency (MAE). In this work, we use the aethalometer measurements of attenuation together with the SMPS-based mass estimation to derive the MAEs of the particles deposited on the filter (l. 262-266). The light absorption at 880 nm was minor, and the 'eBC' concentrations likely biased by the high organic loading. All in all, we expect that the mass of refractory particles was negligible in both the fresh and aged exhaust, compared to the organic aerosol.

*Line 269: "Where PMSMPS is the SMPS-based mass estimate with the assumption that all the mass measured by SMPS is organic, which was a sound assumption considering the AMS results on particle composition." This sentence is unclear. Why not use the AMS-measured organic mass directly, if the assumption is that all mass is organic? Also, how is the assumption justified based on AMS results on particle composition, given that AMS only detects non-refractory species?*

We used the SMPS-derived mass due to the small size of the particles, as the AMS doesn't efficiently capture the sub-50nm particles. For example, in the fresh exhaust the AMS failed to measure the organics altogether due to the small particle size (see Fig. 5 for particle size distributions).

We have edited the sentence on l. 265-266 to better indicate the dominant role of organics to the total particulate emission:

> "This was considered a sound assumption based on the low eBC concentrations and AMS results on non-refractory particle composition."

*Line 450: For the aged emissions, the authors reported EIs of particle mass for the DR 50 experiments.*

*What are the corresponding values for the DR 200 conditions? Would the higher dilution ratio be more atmospherically relevant?*

Particle mass EIs for the DR200 experiments are available in Figure 6. The EIs were similar to the DR50 experiments around 2 equivalent days, and did not increase by elongated photochemical processing. This is now mentioned in text on lines 456-457:

> "The formed particle mass was, however, only slightly higher for the DR50 experiments (Fig. 6a) at similar degree of exposure. Particle mass EI were not found to increase by enhanced photochemical aging after 2 eqv.d."

The choice of dilution ratio is a crucial parameter, but often a compromise especially when extended aging times are considered. In reality, the exhaust plume would be gradually diluted within its atmospheric residence, but this cannot be considered in our experiments due to obvious technical restrictions. The results from two dilution ratios show that dilution does matter when particle composition and amounts are considered, but that the absolute mass-EIs and optical properties of the forming particles might still be close to each other.

*Line 568: Could the authors elaborate on the large difference in BC EIs between fresh and aged exhaust emissions? Can the later-mentioned loading effects fully account for this discrepancy?*

Elemental carbon structures will not form via photochemical processing of organic vapors. Organic coatings can enhance light absorption at 880 nm (Lack and Cappa 2010), which is interpreted as increased 'eBC' when a constant mass-absorption coefficient is applied. Also purely scattering particles can enhance the attenuation when the loading is excessive (Collaud Coen et al. 2010, Virkkula et al. 2015). This is now explained on l. 570:

> "For the aged exhaust measurements, the increase of aerosol scattering due to enhanced OA concentrations likely resulted in positive bias in assessing light absorption from attenuation at 880 nm."

We cannot discern whether the discrepancy between fresh and aged BCs arises purely from the Aethalometer filter loading effects, or is some non-soluble but absorbing particulate matter also formed.

References

Collaud Coen, M., Weingartner, E., Apituley, A., Ceburnis, D., Fierz-Schmidhauser, R., Flentje, H., Henzing, J. S., Jennings, S. G., Moerman, M., Petzold, A., Schmid, O., and Baltensperger, U.: Minimizing light absorption measurement artifacts of the Aethalometer: evaluation of five correction algorithms, Atmos. Meas. Tech., 3, 457–474, https://doi.org/10.5194/amt-3-457-2010, 2010.

Lack, D. A. and Cappa, C. D.: Impact of brown and clear carbon on light absorption enhancement, single scatter albedo and absorption wavelength dependence of black carbon, Atmos. Chem. Phys., 10, 4207–4220, https://doi.org/10.5194/acp-10-4207-2010, 2010.

Peng, Z., and Jimenez, J. L.: Modeling of the chemistry in oxidation flow reactors with high initial NO. Atmos. Chem. Phys, 17(19), 11991–12010. https://doi.org/10.5194/acp-17-11991-2017, 2017.

Peng, Z., and Jimenez, J. L.: Radical chemistry in oxidation flow reactors for atmospheric chemistry research. In Chemical Society Reviews (Vol. 49, Issue 9, pp. 2570–2616). Royal Society of Chemistry. https://doi.org/10.1039/c9cs00766k, 2020.

Virkkula, A., Chi, X., Ding, A., Shen, Y., Nie, W., Qi, X., Zheng, L., Huang, X., Xie, Y., Wang, J., Petäjä, T., and Kulmala, M.: On the interpretation of the loading correction of the aethalometer, Atmos. Meas. Tech., 8, 4415–4427, https://doi.org/10.5194/amt-8-4415-2015, 2015.